# Examining the reliability and validity of two versions of the Effort-Expenditure for Rewards Task (EEfRT)

**Hanno Andreas Ohmann** [1]*, **Niclas Kuper**[2], **Jan Wacker**[1]

1 Faculty of Psychology and Human Movement Science, Universität Hamburg, Hamburg, Germany,
2 Faculty of Psychology and Sports Science, Universität Bielefeld, Bielefeld, Germany

* hanno_ohmann@hotmail.com

## Abstract

The Effort-Expenditure for Rewards Task (EEfRT) has gained validity evidence from several studies. However, various modifications have been applied to the original version, which have never been compared systematically. In Study 1, we tested 120 healthy participants to directly compare two versions of the EEfRT. In Study 2, we tested a larger sample of 394 healthy participants to further examine the original EEfRT. We replicated the split-half reliability of both task versions. However, self-reported personality traits (e.g., trait BAS) correlated with only some task performance parameters in Study 1, which did not replicate for the original EEfRT in Study 2. Our results indicate complex and sometimes inconsistent relations between different personality traits, task properties, and reward attributes.

## 1. Introduction

Harmon-Jones et al. [1] (p. 291) define approach motivation as "the impulse to go toward", which is based on internal state or trait—related processes and initiates behavior which is often (but not necessarily) associated with reaching specific goals. Approach motivation can therefore be seen as one of the main drivers of human behavior [2, 3]. The field of personality research related to approach motivation is often linked to Gray's model of personality, which culminated in his Reinforcement Sensitivity Theory (RST) [4, 5]. A variety of self-report measures are available for the assessment of personality traits related to sensitivity to rewards and punishment, ranging from the commonly used BIS / BAS scales [6] to the Temporal Experience of Pleasure Scale (TEPS) [7] measuring anticipatory pleasure and consummatory pleasure.

However, linking self-reported personality traits to behavioral measurements often leads to rather small correlations [8, 9]. Dang et al. [10] suggest that this might be a result of the poor reliability of many behavioral measures and the different response processes of these two types of measurement. Moreover, multiple behavioral tasks to measure approach motivation have been developed in a post-hoc manner and / or are based on animal models [11]. A majority of these tasks utilize physical effort, which participants have to invest to gain rewards. Various kinds of physical efforts have been utilized. For example, hand-grip tasks assess willingness to

**Competing interests:** The authors have declared that no competing interests exist.

expend effort by force exerted on the grip [12–14]. Such tasks are often adapted to a model based on effort discounting, which allows one to measure the extent to which the need for effort reduces preference for a given reward [15–17]. Similar tasks use button or lever pressing in a progressive ratio format [18–20]. Unfortunately, the majority of these tasks lack a comprehensive test of their reliability and validity. Comparing the psychometric properties of five different effort-based decision-making tasks in a clinical sample (schizophrenic patients) and a healthy control sample, the Effort Expenditure for Rewards Task (EEfRT) [21] exceeded the reliability of four other tasks [14, 22], making it a promising tool to investigate approach motivation.

The EEfRT is based on a concurrent choice paradigm developed by Salamone et al. [11] to explore effort-based decision making in rodents. The original EEfRT measures individual differences in human reward motivation by having participants decide between high cost/high reward (hard task, many clicks needed) and low cost/low reward (easy task, few clicks needed) behavioral options. The tendency to choose the hard task rather than the easy task has been shown to be associated with higher levels of approach motivation, as measured e.g., via personality trait questionnaires [23]. Specifically, trait BAS and trait anticipatory pleasure correlated positively with the percentage of hard-task-choices in trials with a low probability of rewards attainment in the original EEfRT. This indicates that higher approach motivation as measured by trait questionnaires is related to concrete behavior directed at gaining rewards in the EEfRT. Recently, these findings were replicated using a modified version of the EEfRT, showing that trait extraversion and trait BAS correlated positively with the mean number of clicks participants exerted in trials with a low probability of reward attainment [24]. However, Horan et al. [22], Anand et al. [25], as well as Kaack et al. [26] were not able to replicate these correlations using the original EEfRT, raising questions about the existence and magnitude of such associations.

According to Smillie [27], trait behavioral activation system (BAS) sensitivity should be predominantly related to reward sensitivity, while trait behavioral inhibition system (BIS) sensitivity should be predominantly related to punishment sensitivity. One would thus expect associations between behavior in the EEfRT and these personality traits. In particular, trait BAS should be associated with increased task performance (due to greater sensitivity to rewards) and trait BIS with decreased task performance (due to higher sensitivity to the aversive quality of effort expenditure). The EEfRT has gained further support for its validity from various studies (see Table 1). For instance, healthy participants' preference for the hard task has been shown to correlate with lower scores on negative affect, depressive symptoms, and anhedonia [21]. Furthermore, left frontal anodal transcranial direct current stimulation (tDCS) over the dorsolateral prefrontal cortex increased participants' willingness to choose the hard task depending on reward attributes [28], which is in line with models associating left frontal brain activity with approach motivation [29, 30]. Wardle et al. [31] were the first to show that the EEfRT is also sensitive to pharmacological manipulation of dopamine (DA), as d-amphetamine increased participants' overall effort allocation. Furthermore, a low dosage of the D2 receptor blocker sulpiride, which is e.g., used in patients with depression and is believed to increase approach motivation, decreased participants' willingness to exert clicks in a modified version of the EEfRT [24].

Additional evidence for the validity of the EEfRT comes from patients suffering from impaired approach motivation: Patients with schizophrenia [32–34], first-episode psychosis [35], depression [36, 37], and autism [38] were less willing to choose the hard task as compared to healthy controls. Furthermore, the number of hard-task-choices was found to be negatively correlated with the severity of anhedonic symptoms in patients with schizophrenia [33] as well as in patients with depression [37]. Nguyen et al. [39] found in a large healthy sample of

**Table 1. An overview of selected studies using the EEfRT: Modifications, reliability, and validity (chronological order).**

| Study | N | Main dependent variable(s) | Main modifications applied to the original task | Modifications tested against original? | Reliability reported? | Validity test |
|---|---|---|---|---|---|---|
| Treadway et al. 2009 [21] | 61 | HTC | - | - | No | Self-report measures (Anhedonia, depression) |
| Wardle et al. 2011 [31] | 17 | HTC | Repeated-measure-design | No | No | Effects of d-amphetamine |
| Damiano et al. 2012 [38] | 58 | HTC, response change | Removed time-limit for task choice | No | No | Patients with ASD (n = 20) VS healthy controls (n = 38) |
| Wardle et al. 2012 [45] | 23 | HTC, response speed | Practice session, repeated measure | No | No | Effects of caffeine |
| Fervaha et al. 2013 [32] | 32 | HTC | Individual calibration of required clicks | No | No | Patients with SCZ (n = 16) VS healthy controls (n = 16) |
| Barch et al. 2014 [33] | 98 | HTC | Removed low probability trials | No | No | Patients with SCZ (n = 59) VS healthy controls (n = 39); self-report measures (anhedonia, depression) |
| Yang et al. 2014 [37] • study 1 | 99 | HTC | Reduced number of required clicks; fixed the possible monetary rewards | No | No | high BDI-score (n = 43) VS low BDI score (n = 56); self-report measures (anhedonia; pleasure) |
| • study 2 | 87 | HTC | Reduced number of required clicks; fixed the possible monetary rewards | No | No | Patients with MDD (n = 41) VS remitted MDD (n = 41) |
| Hughes et al. 2015 [46] | 51 | HTC | Paid 10% of total winnings | No | No | Association with frontal asymmetry (resting state) |
| Geaney et al. 2015 [23] | 97 | HTC | Mood induction; paid 10% of total winnings | No | No | Self-report measures (anhedonia, pleasure, BAS) |
| Gilman et al. 2015 [47] | 50 | HTC; reaction time | Pictures of peers (social influence) | No | No | Effects of different social influence conditions |
| Reddy et al. 2015 [14] | 134 | Reward sensitivity | Repeated-measure-design; individual calibration of required clicks; no low probability trials | No | **Yes (retest)** | Patients with SCZ (n = 94) VS healthy controls (n = 40); comparison with four other tasks |
| Horan et al. 2015 [22] | 134 | Reward sensitivity | Repeated-measure-design; individual calibration of required clicks; no low probability trials | No | **Yes (retest)** | Self-report measures (SCZ symptoms; motivation; BIS/BAS) |
| Hughes et al. 2017 [48] | 128 | HTC | Repeated-measure-design; shorter task selection (3s) | No | No | Abstinent smokers (n = 61) VS former smokers (n = 67); self-reported reward enjoyment |
| Johnson et al. 2017 [49] | 50 | HTC | Fixed probability (50%), three bonus trials added (no effort needed) | No | No | Patients with remitted bipolar disorder; self-reported life ambitions |
| Lopez-Gamundi & Wardle 2018 [51] | 60 | HTC | Cognitive effort version | **Yes** | No | Comparison of cognitive VS original version |
| Ohmann et al. 2018 [28] | 60 | HTC | Repeated-measure-design; paid 5% of the total winnings | No | No | Effects of anodal tDCS stimulation |
| Racine et al. 2019 [50] | 63 | HTC | Food portions instead of monetary reward | No | No | Comparison of participants with different degree of binge-eating symptoms; BMI |
| Nguyen et al. 2019 [39] | 2259 | Reward sensitivity | No probabilities of reward attainment for kids | No | No | Comparison of parents (n = 1044) and children (n = 1215) for psychopathic symptoms |
| Byrne & Ghaiumy Anaraky 2019 [52] | 94 | HTC | Addition of "loose" trials; temporal delay after easy trials | No | No | Comparison of older (n = 46) and younger adults (n = 48) |
| Ohmann et al. 2020 [24] | 203 | Button presses | No task selection, instead increased potential monetary gains with each click | No | **Yes (split-half)** | Effects of sulpiride; self-report measures (Big Five; BIS/BAS) |

(*Continued*)

**Table 1.** (Continued)

| Study | N | Main dependent variable(s) | Main modifications applied to the original task | Modifications tested against original? | Reliability reported? | Validity test |
|-------|---|---------------------------|-------------------------------------------------|----------------------------------------|----------------------|---------------|
| Kaack et al. 2020 [26] | 49 | HTC | Additional "offer"- screen (5s) prior to choice screen; best performing student would receive $100 grocery store voucher | No | No | Associations with frontal asymmetry (resting and task states); self-report measures (BIS/BAS) |

*Note.* ASD = Autism spectrum disorder; BAS = Behavioral Activation System; BIS = Behavioral Inhibition System BDI = Beck Depression Inventory; BMI = Body-mass-index; HTC = Hard-task-choices; MDD = major depressive disorder; SCZ = schizophrenia

parents and their children that symptoms of psychopathology correlated with reduced effort allocation within the EEfRT.

The EEfRT has also gained evidence on a neurophysiological level, as it has been shown to be related to left-frontal cortical asymmetry in the resting state as well as during task performance [26, 40], which is believed to be a neural signature of approach motivation. Moreover, Huang et al. [41] found that the percentage of hard-task-choices was directly linked to the activity of the NAcc, which is a key structure of the human reward circuit, in both patients with schizophrenia and healthy participants. Overall, the literature thus shows intriguing support for the validity of the EEfRT.

However, there are also various limiting aspects (see Table 1). First, the number of studies reporting a significant link between the behavioral measurements within the EEfRT and self-reported personality traits related to approach motivation is still small, although many studies refer to this link as validity evidence of the EEfRT. Second, the number of participants in studies which used the EEfRT has often been relatively small, resulting in low statistical power to detect effects sizes that can be expected in individual difference research [42]. Together with concerns about the replicability of psychological findings in general [43] as well as literatures relevant to this manuscript (e.g., trait approach motivation–frontal asymmetry link) [44], this highlights the risk of false positive results in previous studies. Third, the original EEfRT has been shown to be partly related to individual strategic behavior, which is not related to participants' actual approach motivation [28]. The modified version of the EEfRT [24] seeks to eliminate this limitation of the original task, but requires additional data to document its reliability and validity. Fourth, seemingly small differences in task properties and administration could have a great impact on task behavior. Despite this, several studies already modified the original EEfRT to fit different experimental settings (see Table 1).

Just to name a few examples: Yang et al. [37] reduced the number of required clicks and fixed the possible monetary rewards to reduce the complexity of the task for depressive patients. This design was also used by Huang et al. [41] to make the EEfRT suitable for functional brain imaging. Barch et al. [33] completely removed the low probability of reward attainment category, Damiano et al. [38] removed the time limit when participants select either the easy or the hard task, Fervaha et al. [32] calculated an individual number of required clicks before the actual task based on motoric abilities, and Byrne & Ghaiumy Anaraky [52] introduced "loss trials", in which choosing the easy task leads to potentially higher monetary loss compared to the hard task. Other authors exchanged the monetary rewards of the EEfRT, e.g., by using food portions in a study with patients suffering from binge-eating [50]. A different study added a social influence aspect while participants chose between the easy and hard task of the EEfRT by adding pictures of "peers" and their respective choices [47]. Despite these various modifications applied to the EEfRT so far, surprisingly little is known about the effect of such modifications as almost no study directly compared different versions of the EEfRT

within one experimental design. A commendable exception is a study by Lopez-Gamundi & Wardle [51], who compared the original EEfRT to a modification which uses cognitive effort (set-switching-task; C-EEfRT) instead of the physical effort within the original task (clicks). Although participants perceived the C-EEfRT as more difficult, participants did choose the hard task more often compared to the original EEfRT. Furthermore, the relationship between the effort allocation within both task versions was only moderate, indicating distinct processes for both kinds of effort when participants decide to allocate effort to gain a possible reward.

## 1.1 Present study

Bearing these limitations in mind, we here seek to analyze the reliability and validity of the original EEfRT and a modified EEfRT in two different study designs and try to deepen the understanding of the link between self-reported personality traits and behavioral task measures. Study 1 aims at directly comparing the validity and reliability of two versions of the EEfRT. Study 2 aims to replicate the reliability and validity of the original EEfRT by making use of a large sample to further increase statistical power.

**1.1.1 Reliability of the EEfRT.** Based on the promising results regarding the retest-reliability found in previous studies for the original EEfRT [14, 22] and likewise promising results for the split-half reliability of the modified EEfRT [24], we expected both versions of the task to show overall good (Rel > .80) split-half reliability in both studies. We further examined the internal consistency of all questionnaire measures used in both studies.

**1.1.2 Validity of basic task variables.** In line with previous research, we expected the reward attributes (reward magnitude and probability of reward attainment) to be positive predictors of the percentage of hard-task-choices (original EEfRT) and mean number of clicks (modified EEfRT), whereas we expected trial number (i.e., an indicator of fatigue) to be a negative predictor of the percentage of hard-task-choices (original EEfRT) and mean number of clicks (modified EEfRT). GEE models (generalized estimating equations) [53, 54] have been the main analysis strategy for the examination of basic task variables in previous work on the EEfRT [21, 23, 24]. Therefore, we also applied GEE models to test for the effects of the above-mentioned basic task variables.

**1.1.3 Personality correlations.** As Gignac & Szodorai [42] stated, correlations of $r = .30$ should be considered as rather large correlations in the field of individual differences. Applying these standards, previous studies using the original EEfRT [23] as well as the modified version of the EEfRT [24] found medium to large correlations for the percentage of hard-task-choices (original EEfRT) and the mean number of clicks (modified EEfRT) in trials with low probability of reward attainment with trait BAS ($r$ ranging from .212 to .361) and trait anticipatory pleasure ($r = .251$). We seek to replicate these correlations in both studies. Furthermore, as some studies using the EEfRT focused on the impact of reward magnitude, as well as the differences between low and high reward trials [14, 22], indicating "reward sensitivity", we exploratorily analyzed the correlations between traits with the percentage of hard-task-choices (original EEfRT) and mean number of clicks (modified EEfRT) depending on reward magnitude. Therefore, we examined the correlations between trait measures and task performance in all probability of reward attainment categories as well as in all reward magnitude categories and difference scores between these trial categories in an exploratory fashion. To evaluate the discriminant validity of both tasks, we further tested for the associations with distinct constructs, which we expected to not correlate with effort allocation on both tasks—namely risk-taking (behavioral measure) and impulsivity (self-reported trait measure).

**1.1.4 Secondary analysis.** As Study 1 includes both versions of the EEfRT in one experimental design, we seek to estimate the correlations between both task versions in an

exploratory fashion. As there are no previous studies to base our hypotheses on, we only hypothesize a significant positive correlation between both versions, as both tasks should measure the same construct: approach motivation. We further analyzed participants' self-reported strategy usage and motivation and their linkage to task performance in Study 1 to further test the validity of both task versions.

## 2. Methods

### 2.1 Study 1

**2.1.1 Participants.** We recruited physically and psychologically healthy participants (78.3% female) aged 18–35 ($M$ = 24.97; $SD$ = 4.14) using online notice boards and flyers at a local university. Out of 125 recruited participants, 5 had to be excluded for different reasons (1 participant was not able to understand the instructions; 2 participants did not understand the task; 2 participants had missing values due to technical failure). Thus, the final sample consisted of 120 participants. In line with previous studies [23, 24], we expected correlations around $\rho$ = .30 (i.e., large correlations according to Gignac & Szodorai [42]). As intended, statistical power was therefore >.80 (exact 1-β = .92) to detect correlations of $\rho$ = .30 ($\alpha$ = .05). Participants received monetary compensation (10€ per hour) and were told that they could gain additional money based on their collected rewards from both versions of the EEfRT (5% of the virtually collected money) and the BART (one cent per 4 pumps in successful trials) which was paid to participants at the end of the study. The authors assert that all procedures contributing to this work comply with the ethical standards of the relevant national and institutional committees on human experimentation and with the Helsinki Declaration of 1975, as revised in 2008. The study has been approved by the local Ethics Committee of the University of Hamburg. Exclusion criteria comprised the intake of any kind of prescribed medication over the last three months, the consumption of illegal drugs over the last four weeks, neurological or medical conditions, and the presence of any mental disorders (in particular affective, somatoform, psychotic, anxiety, eating, and adaptive disorders, as well as substance use disorders).

**2.1.2 Randomization.** As we intended to compare both versions of the EEfRT (original / modified) and their relation to measures of personality traits and risk-taking behavior, we randomized the order of both tasks in a counterbalanced fashion to ensure that the order of both tasks did not influence effort allocation. Participants were randomly assigned to one of both conditions at the start of part two of our study.

**2.1.3 Procedure.** Participants who fit the inclusion criteria were provided with information about the study via email. Participants gave their informed consent in written form via email and then received an individual link to the first part of the study. The first part of the study comprised an online-survey lasting about 30 minutes, in which participants filled out a series of questionnaires, including demographic information and German versions of the BIS/ BAS scales [6, 55], the TEPS [7], and the UPPS [56, 57]. Participants who completed the first part of the study were invited into our lab. After arriving at our lab, participants personally signed the informed consent which they sent before via E-Mail. Afterwards, the participants completed a series of computer tasks lasting about 60 minutes. All participants started by completing a test of their motoric abilities. Participants then completed either the original or the modified version of the EEfRT according to the assigned condition. Afterwards, they completed the Balloon Analogue Risk Task (BART) [58, 59]. The BART was always completed in the middle of the study to avoid motoric fatigue of the participants throughout the whole study. The BART was followed by another test of motoric abilities and the complementary version of the EEfRT, which participants did not complete before. Finally, participants completed

a small set of questions, including questions about aspects that might have influenced their effort allocation within each version of the task (reward magnitude, probability of reward, fatigue, resting one's fingers) and their motivation to earn additional money throughout the whole study using five-point Likert-scales ranging from "not at all" (1) to "a lot" (5).

**2.1.4 Original EEfRT.** We used a translated (German) version of the EEfRT [21], which was programmed using Presentation software 17.1 (Neurobehavioral Systems Inc, San Francisco). Every participant completed one 15–minute block of the EEfRT with their dominant hand. Participants were instructed to win as much virtual money as possible throughout the block. In short, participants need to choose between an easy, low-reward task and a hard, high-reward task in every trial (see Fig 1 for a schematic illustration). The reward for the easy task is fixed to 1 € while the reward for the hard task is variable (ranging between 1.21 € and 4.30 €). To further manipulate the value of each reward, the probability of reward attainment also varies [either 12% (low), 50% (medium) or 88% (high)], which is presented at the start of each trial alongside the reward values. The easy task requires participants to press the space button 30 times ("clicks") in 7 seconds with their index finger. The hard task requires participants to press the space button 100 times ("clicks") in 21 seconds with their pinkie finger. While pressing the spacebar, a visually presented white bar gradually fills up with red color.

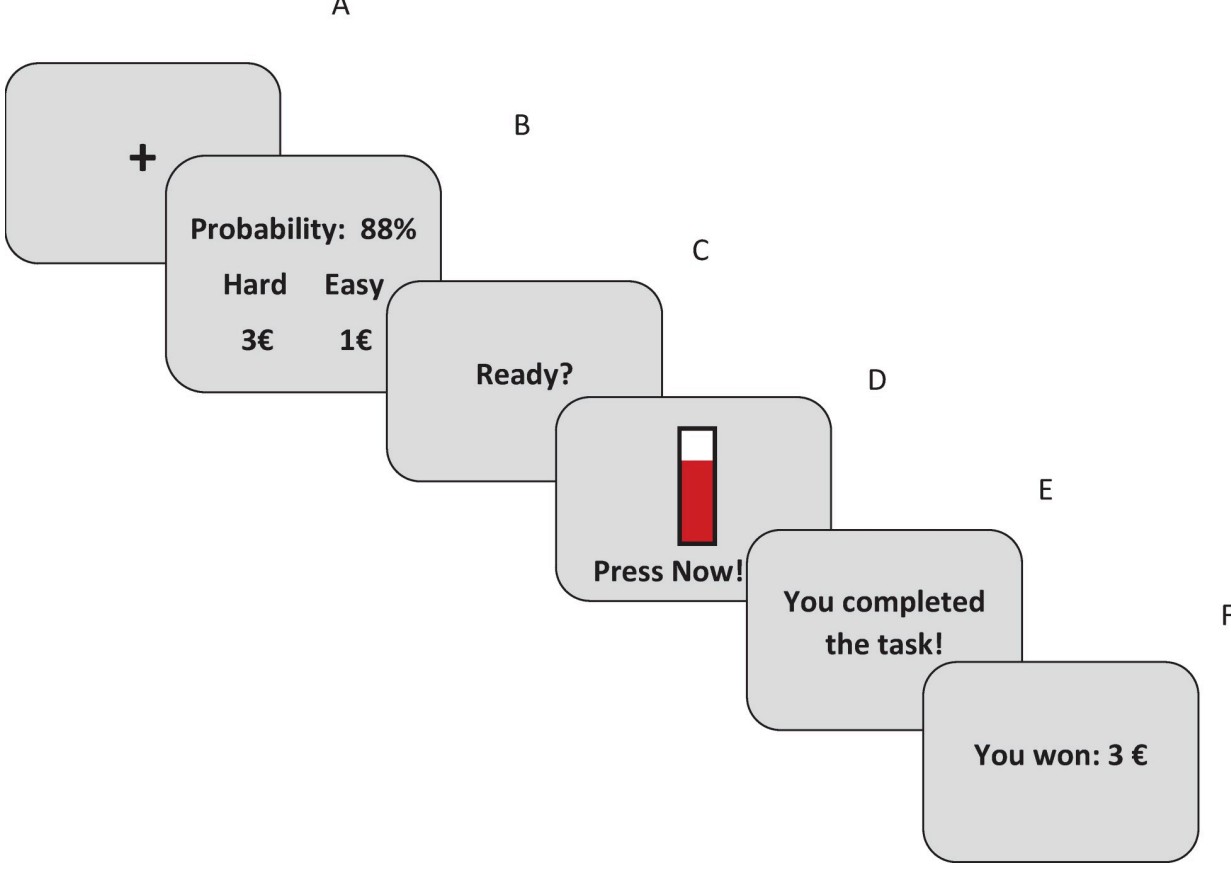

**Fig 1. Schematic illustration of one trial of the original EEfRT.** A fixation cross (1s, A) is followed by a screen showing probability of reward attainment and reward magnitude for the easy and the hard task (B), lasting until the participant made a choice which task to complete but no longer than 5s. Then, after presentation of a ready–screen (1s, C) the main screen for the trial showing a red bar that fills with each button press is presented until the task is completed or until the trial time is over (D). Finally, task completion is signaled (2s, E) and a feedback screen shows the amount of money won (2s, F).

After each trial, the participants are informed about the amount of money they won during the trial. The order of trials, as well as the probability of reward attainment and the reward magnitudes are not randomized between participants, but pre-assigned for each trial. This is done to rule out random feedback differences between participants.

**2.1.5 BART.** To test participants'risk-taking behavior, we also assessed the BART [58, 59]. In this task, participants are presented a picture of a balloon, and instructed to inflate this balloon. Inflating the balloon increases the size of the balloon on screen and the associated reward. However, overinflating the balloon would result in the balloon bursting accompanied by an aversive auditory sound. Bursting of the balloon causes the participant to lose the entire reward of that trial. Each of 30 balloons have a different predetermined bursting point, on a scale of 1 to 128 (pumps). Participants are instructed that the average number of inflations that causes the balloon to burst is 64 and that they would gain one Cent per 4 pumps, only in successful trials (balloon did not burst). In line with previous studies, we used the automatic response procedure of the BART [59, 60], in which the participants could immediately select the intended number of pumps for that specific trial and receive immediate feedback as they watch the balloon inflating. Also, risk taking scores were calculated as the mean of the number of pumps across all balloons regardless of the burst event [59], unlike in the original BART [58].

**2.1.6 Modified EEfRT.** Additionally, we used a modified version of the original EEfRT [21], which was programmed using Presentation software 17.1 (Neurobehavioral Systems Inc, San Francisco) and has been used in one previous study [24]. Ohmann et al. [28] found that using the original EEfRT comes with a major downside: At least some participants understand that choosing the hard task is often lowering the possible overall monetary gain as the hard task takes almost 3 times as long as the easy task and the overall duration of the task is fixed. Hence, at least some participants' choices are partly based on a strategic decision and less on approach motivation per se. To overcome this downside, the original EEfRT was modified substantially. First, the number of trials (2 blocks x 15 trials = 30 trials) and the duration of each trial (= 20 seconds) was fixed. Participants used their dominant hand for both blocks in the present study. Second, the original choice-paradigm was changed. Participants no longer choose between an easy and a hard task. As in the original task, the value of each reward varies, and participants are informed about this at the start of each trial. But instead of presenting specific reward magnitudes, participants are now presented with a reward magnitude per click (1 /2 / 3 / 4 / 5 cents per click). Thus, participants are able to increase the total possible monetary gain in each trial with each click. In accordance with the original task design, the probability of reward attainment also varied [either 12% (low), 50% (medium) or 88% (high)], which is presented at the start of each trial alongside the reward value per click. Participants were instructed to win as much virtual money as possible throughout the task, however they were free to choose the amount of effort they exerted in each trial. Critically, the only way to increase the possible monetary gain is to increase the number of clicks in each trial. The task itself is designed to be close to the original EEfRT but comes with some modifications to prevent the use of strategies (see Fig 2). While pressing the spacebar, a visually presented red bar gradually grows. A scale (€) was implemented, so that the participants can always see how much their button-presses ("clicks") increase their possible monetary gain. Furthermore, the information on the reward magnitude per click and the probability of reward attainment is presented throughout the whole trial alongside a countdown (20 seconds) to increase participants' awareness of these parameters. After each trial, participants are informed about the amount of money they won during the trial. The order of trials, as well as the probability of reward attainment and the reward magnitudes per click are not randomized between participants, but pre-assigned for each trial. This is done to rule out random feedback differences between participants.

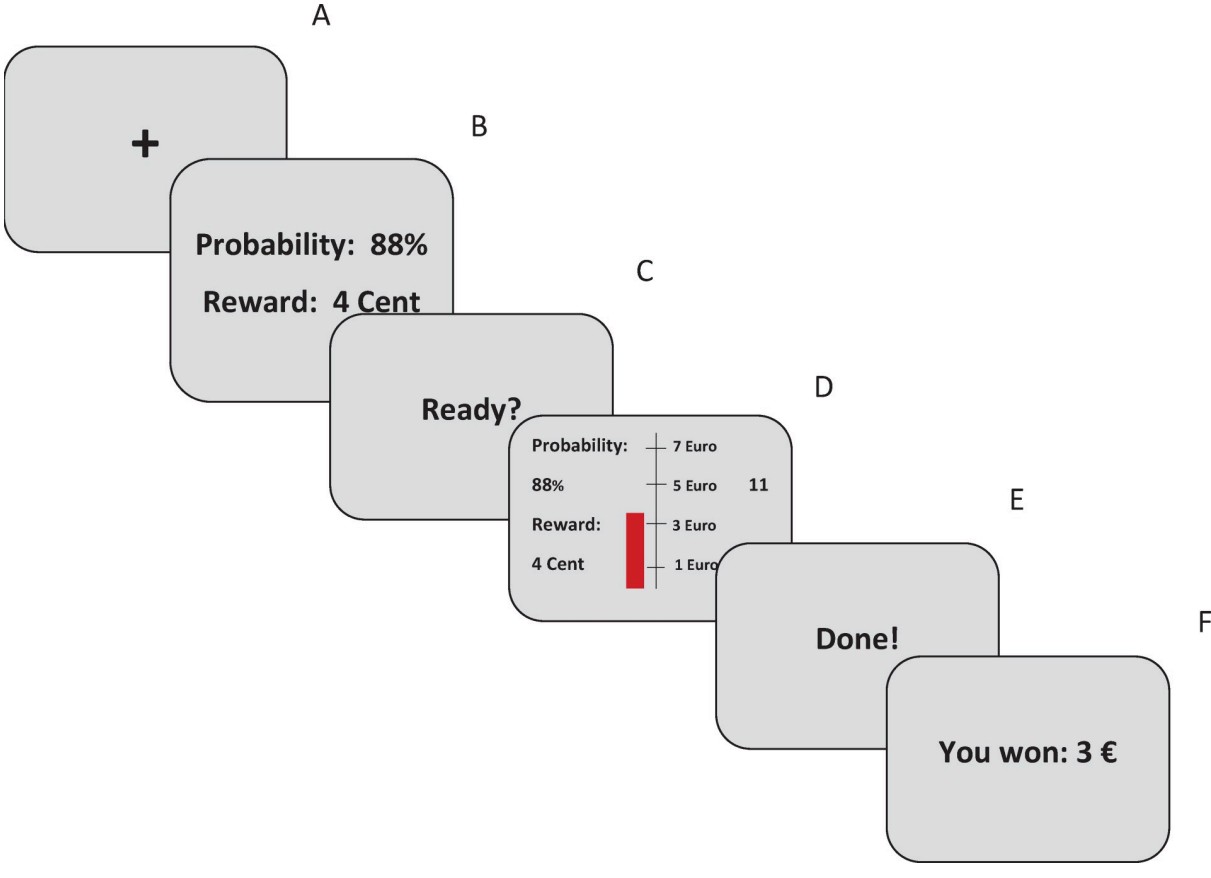

**Fig 2. Schematic illustration of one trial of the modified EEfRT.** A fixation cross (1s, A) is followed by a screen showing probability of reward attainment and reward magnitude per click for 3s (B). Then, after presentation of a ready–screen (1s, C), the main screen for the trial showing a red bar that grows with each click is presented alongside a scale, indicating the current monetary gain and a countdown (20s, D). Finally, task completion is signaled (1.5s, E) and a feedback screen shows the amount of money won (2s, F).

**2.1.7 Motoric abilities.** Participants with greater motoric ability exert more clicks throughout the modified version of the EEfRT [24] and studies calibrating an individual number of clicks to succeed within the original EEfRT suggest that participants with higher motoric abilities might also choose the hard task more often in the original version [14, 32], which does not reflect their actual approach motivation. Therefore, we included 10 motoric trials to test participants' motoric abilities before each version of the EEfRT. Within these motoric trials, participants were instructed to press the spacebar as often as possible within 20 seconds. Critically, participants were not able to gain any rewards in these trials and visual feedback was reduced to a countdown and a display of the number of clicks they exerted. Participants' individual motoric abilities were operationalized as maximal clicks in motoric trials (MaxMot) and included in our statistical models.

**2.1.8 Data analysis.** Aggregated data were analyzed using the SPSS 26.0 software* (Chicago, IL, USA). First, we examined the split-half reliability of different task parameters. To this end, we split the dataset into the first and second temporal half of trials for each individual in the original task. In the modified task, split-half reliability was estimated across the two blocks. Furthermore, we calculated internal consistencies of the questionnaire measures used.

To test for the effects of basic predictors on the percentage of hard-task-choices (original EEfRT) and on the mean number of clicks (modified EEfRT), we used GEEs. GEEs are

marginal models that allow for robust parameter estimation despite correlated residuals, e.g., due to the clustering of trials within participants [53, 54]. Crucially, GEEs are consistent and provide appropriate robust standard errors even when the correlation matrix for the residuals is specified incorrectly [54]. Models were fit using an exchangeable working correlation matrix. Given that our dependent variable (hard-task-choices) was binary, we implemented models using the binomial distribution with a logit link. For the modified task (dependent variable: number of clicks), a gaussian distribution was assumed. All GEE models included the factors trial number, probability (categorical), reward magnitude, and the interaction of probability x reward magnitude (often referred to as "expected value"). Moreover, participants' individual motoric abilities were included in all GEE models.

Pearson correlations were computed between self-reported personality traits (BIS/BAS/TEPS/UPPS), the number of pumps within the BART, the percentage of hard-task-choices within the original EEfRT, and the mean number of clicks within the modified EEfRT, separately for each probability of reward attainment category (low/medium/high), for each reward magnitude category (low/medium/high), as well as for difference scores between these categories. Within the original EEfRT, we formed these categories in line with previous studies [46]; low reward magnitude: <2,30 Euro, medium reward magnitude: 2,31–3,29 Euro, high reward magnitude: >3,30 Euro. For the modified EEfRT, we calculated analogue categories: Low reward magnitude: 1 or 2 Cent per click, medium reward magnitude: 3 Cent per click, high reward magnitude: 4 or 5 Cent per click. Further Pearson correlations were calculated between both versions of the EEfRT for each probability of reward attainment category, each reward magnitude category, as well as between both versions of the EEfRT and follow-up questions regarding individual strategies and motivation. Datasets and syntax can be found at: https://osf.io/35k2w/.

## 2.2 Study 2

**2.2.1 Participants.** We recruited physically and psychologically healthy, right-handed participants (68.2% female) aged 18–50 ($M$ = 25.74; $SD$ = 5.37) using online notice boards and flyers at various universities. Out of 409 recruited participants, 15 had to be excluded for different reasons (11 participants consumed illegal drugs or hormones within the last 12 months, 1 subject already knew the EEfRT task, 1 subject was ambidextrous,1 subject had missing values for the UPPS, and for 1 subject it was not possible to collect a blood sample). Thus, the final sample consisted of 394 participants. In accordance with Gignac & Szodorai [42], we applied a more conservative criterion than in Study 1 considering the discrepancy between published correlations and those correlations we found in our lab and expected only medium-sized correlations. As intended, statistical power was therefore >.80 (exact 1-β = .98) to detect correlations of $\rho$ = .20 ($\alpha$ = .05). Participants received monetary compensation (10€ per hour) and were told that they could gain additional money based on their collected rewards from the original EEfRT (5% of the virtually collected money). At the end of the study, all participants received an additional 5€ to ensure equity. This amount was always higher than the 5% of the virtually collected money.

The authors assert that all procedures contributing to this work comply with the ethical standards of the relevant national and institutional committees on human experimentation and with the Helsinki Declaration of 1975, as revised in 2008. The study has been approved by the local Ethics Committee of the Medical Chamber of Hamburg. Exclusion criteria comprised the regular intake of any kind of prescribed medication, consumption of illegal drugs over the last 12 months, neurological or medical conditions, and the presence of any mental disorders —in particular affective-, somatoform-, psychotic-, anxiety-, eating-, and adaptive disorders,

as well as substance use disorders, high consumption of alcohol (more than 15 glasses per week), nicotine (more than 15 cigarettes per week) or caffeine (more than 4 cups per day), and pregnancy (tested onside).

**2.2.2 Procedure.**   As the original EEfRT and relevant questionnaires (BIS/BAS, UPPS) were deployed as part of a larger study, we will focus only on the most relevant information. A complete description of the procedure will be provided elsewhere. The order of tasks and questionnaires was not randomized and in contrast to Study 1, no motoric trials were included. Participants who fit the inclusion criteria were invited into the lab once. After arriving at the lab and giving their informed consent in written form, participants completed a large series of questionnaires, including the German BIS/BAS [6, 55] and the UPPS [56, 57], as well as a series of computer tasks, including the original EEfRT [21]. The overall experiment lasted for about 4.5 hours. The EEfRT took place approximately 4 hours after the start of the experiment.

**2.2.3 Original EEfRT.**   The procedure of the original EEfRT was identical to experiment 1 (see 2.1.4 and Fig 1).

**2.2.4 Data analysis.**   Statistical analysis for the original EEfRT was identical to experiment 1 (see 2.1.8), with the following exceptions: As no motoric trials were included, individual motoric abilities were removed as a factor from the GEE model. In accordance with Study 1, Pearson correlations were computed between self-reported personality traits (BIS/BAS/ UPPS) and the percentage of hard-task-choices within the original EEfRT for each probability of reward attainment (low/medium/high) and each category of reward magnitude (low/medium/ high), as well as difference scores between these categories. Datasets and syntax can be found at: https://osf.io/35k2w/.

# 3. Results

## 3.1 Reliability

We examined the internal consistency of all questionnaires in both studies by estimating Cronbach's Alpha. Most questionnaires showed moderate to good internal consistency (see Table 2). We then calculated the reliability of the EEfRT by estimating split-half correlations and applying Spearman-Brown corrections to the resulting estimates. Note however, as we introduced motoric trials within Study 1, we compared reliabilities unadjusted and adjusted for motoric abilities, respectively (see Table 2). Therefore, we calculated the reliability of the percentage of hard-task-choices and clicks after residualizing them on motoric abilities (operationalized as maximal clicks in motoric trials; MaxMot). The overall HTCs and clicks showed high reliability in both studies, within the original EEfRT in Study 1 (Rel = .90; Adj. Rel = .91) and in Study 2 (Rel. = .87), and within the modified EEfRT in Study 1 (Rel = .96; Adj Rel. = .92)

When separately analyzing the data for each probability and reward magnitude category, split-half reliabilities ranged between Rel = .73 and .97 over both studies and task versions. However, when calculating the split-half reliability for the difference scores, some indices showed relatively poor reliability (range: Rel. = .35 to .86), especially the difference between high and medium reward magnitudes. Overall, both versions of the EEfRT showed good reliability and the adjustment for motoric abilities in Study 1 impacted the reliability of the EEfRT only slightly.

## 3.2 Experiment 1

**3.2.1 Original EEfRT—validity of basic task variables.**   Participants on average chose the hard task in 53.41% of all trials ($SD$ = 18.50%; Range = 2.90–100%). We conducted a GEE model to test validity of the basic task variables within the original EEfRT in Study 1 (see

**Table 2. Internal consistencies and reliabilities for questionnaires and both versions of the EEfRT in Study 1 and 2.**

|  | Study 1 | Study 2 |
|---|---|---|
| Questionnaires |  |  |
| BAS | $\alpha = 0.79$ | $\alpha = 0.74$ |
| BIS | $\alpha = 0.78$ | $\alpha = 0.79$ |
| TEPS Anticipatory | $\alpha = 0.67$ | - |
| TEPS Consummatory | $\alpha = 0.70$ | - |
| UPPS Urgency | $\alpha = 0.86$ | $\alpha = 0.84$ |
| UPPS Premeditation | $\alpha = 0.78$ | $\alpha = 0.77$ |
| UPPS Perseverance | $\alpha = 0.83$ | $\alpha = 0.82$ |
| UPPS Sensation Seeking | $\alpha = 0.84$ | $\alpha = 0.83$ |
| Original EEfRT task |  |  |
| HTCs | Rel = .90 / $Rel_{adj}$ = .91 | Rel = .87 |
| Low Probability HTCs | Rel = .84 / $Rel_{adj}$ = .84 | Rel = .81 |
| Medium Probability HTCs | Rel = .79 / $Rel_{adj}$ = .78 | Rel = .80 |
| High Probability HTCs | Rel = .77 / $Rel_{adj}$ = .76 | Rel = .73 |
| High–Low Probability HTCs | Rel = .76 / $Rel_{adj}$ = .76 | Rel = .75 |
| High–Medium Probability HTCs | Rel = .67 / $Rel_{adj}$ = .66 | Rel = .57 |
| Low Reward HTCs | Rel = .82 / $Rel_{adj}$ = .83 | Rel = .80 |
| Medium Reward HTCs | Rel = .82 / $Rel_{adj}$ = .81 | Rel = .78 |
| High Reward HTCs | Rel = .82 / $Rel_{adj}$ = .82 | Rel = .79 |
| High–Low Reward HTCs | Rel = .67 / $Rel_{adj}$ = .67 | Rel = .66 |
| High–Medium Reward HTCs | Rel = .39 / $Rel_{adj}$ = .38 | Rel = .40 |
| Modified EEfRT task |  |  |
| Clicks | Rel = .96 / $Rel_{adj}$ = .92 | - |
| Low Probability Clicks | Rel = .91 / $Rel_{adj}$ = .87 | - |
| Medium Probability Clicks | Rel = .96 / $Rel_{adj}$ = .91 | - |
| High Probability Clicks | Rel = .97 / $Rel_{adj}$ = .93 | - |
| High–Low Probability Clicks | Rel = .86 / $Rel_{adj}$ = .86 |  |
| High–Medium Probability Clicks | Rel = .68 / $Rel_{adj}$ = .68 |  |
| Low Reward Clicks | Rel = .93 / $Rel_{adj}$ = .89 | - |
| Medium Reward Clicks | Rel = .95 / $Rel_{adj}$ = .89 | - |
| High Reward Clicks | Rel = .95 / $Rel_{adj}$ = .89 | - |
| High–Low Reward Clicks | Rel = .80 / $Rel_{adj}$ = .80 | - |
| High–Medium Reward Clicks | Rel = .37 / $Rel_{adj}$ = .35 | - |

*Note.* Depicted are internal consistencies of questionnaire measures and split-half reliabilities of task measures. = Cronbach's Alpha, Rel = split-half reliability calculated by Spearman-Brown correcting the split-half correlation of task measures (derived by splitting the dataset into the first and second temporal half of trials for each individual in the original task; calculated across the two blocks in the modified task), adj = adjusted for motor abilities: variables were predicted by the maximum number of clicks in motoric trials and residuals were used to calculate reliability, EEfRT = Effort Expenditure for Reward Task, HTC = Hard Task Choices.

Table 3). The GEE model examined the main effects of task-dependent variables (reward magnitude, probability of reward attainment, and trial number) as well as one variable unique to our study (MaxMot) on the percentage of hard-task-choices. In line with previous studies, a significant positive main effect was found for reward magnitude and probability of reward attainment and a significant negative main effect was found for trial number, indicating that all three factors were predictors of percentage of hard-task choices (all $p$s < .001).

**Table 3. GEE models for basic predictors of percentage of hard-task choices within the original EEfRT in Study 1 and 2 and of mean number of clicks within the modified EEfRT in Study 1.**

| Effect | B | se | $\chi^2$ | p |
|---|---|---|---|---|
| **Study 1 –Original EEfRT** | | | | |
| Reward Magnitude | 0.80 | 0.08 | 111.38 | **< .001** |
| Probability 88%[a] | 3.44 | 0.28 | 146.59 | **< .001** |
| Probability 50%[a] | 1.59 | 0.13 | 157.97 | **< .001** |
| Probability 88%[a] × Reward Magnitude | 1.32 | 0.20 | 44.46 | **< .001** |
| Probability 50%[a] × Reward Magnitude | 0.51 | 0.10 | 27.68 | **< .001** |
| Trial | -0.77 | 0.12 | 41.81 | **< .001** |
| MaxMot | 0.00 | 0.01 | 0.08 | .772 |
| **Study 1 –Modified EEfRT** | | | | |
| Reward Magnitude | 4.91 | 0.46 | 111.80 | **< .001** |
| Probability 88%[a] | 17.80 | 1.53 | 135.98 | **< .001** |
| Probability 50%[a] | 12.47 | 1.07 | 135.35 | **< .001** |
| Probability 88%[a] × Reward Magnitude | -1.83 | 0.43 | 18.51 | **< .001** |
| Probability 50%[a] × Reward Magnitude | -1.33 | 0.38 | 12.19 | **< .001** |
| Trial | -5.96 | 0.90 | 43.58 | **< .001** |
| Block | -2.22 | 0.57 | 14.95 | **< .001** |
| MaxMot | 0.61 | 0.05 | 149.48 | **< .001** |
| **Study 2 –Original EEfRT** | | | | |
| Reward Magnitude | 0.66 | 0.04 | 299.55 | **< .001** |
| Probability 88%[a] | 3.10 | 0.14 | 508.21 | **< .001** |
| Probability 50%[a] | 1.60 | 0.07 | 587.79 | **< .001** |
| Probability 88%[a] × Reward Magnitude | 0.91 | 0.10 | 90.63 | **< .001** |
| Probability 50%[a] × Reward Magnitude | 0.40 | 0.05 | 65.19 | **< .001** |
| Trial | -0.72 | 0.07 | 103.37 | **< .001** |

*Note*. All models included probability (categorical), reward magnitude, and trial number (divided by the individual maximum number of trials) as within-subjects variables; $\chi^2$ = Wald chi-square; $B$ = regression coefficient; significant effects in **bold**.

[a]Reference category: 12% probability

Furthermore, the interaction of reward and probability (often referred to as "expected value") did also reach significance, indicating that higher probability did predict a higher percentage of hard-task-choices with increasing reward magnitude (see Fig 3). The factor MaxMot did not reach significance ($B$ = 0.00, $\chi^2(1)$ = 0.08, $p$ = .772), indicating that motoric ability as measured within the motoric trials did not strongly affect hard task choices within the original EEfRT.

**3.2.2 Modified EEfRT—validity of basic task variables.** Participants on average exerted 117.71 number of clicks in each trial ($SD$ = 16.03; Range = 77.17–156.43). In accordance with our analysis of the original EEfRT (see 3.1.1), we computed a GEE model to test validity of the basic task variables within the modified EEfRT in Study 1 (see Table 3). The GEE Model examined main effects of task-dependent variables (reward magnitude, probability of reward attainment, and trial number) as well as one variable unique to our study (MaxMot) capturing the mean number of clicks. In line with previous studies, significant positive main effects were found for reward magnitude and probability of reward attainment and a significant negative main effect was found for trial number, indicating all three factors were predictors of the mean number of clicks (all $p$s < .001). Furthermore, the interaction of reward and probability also reached significance ($p$ < .001), indicating that higher probability predicted a stronger increase

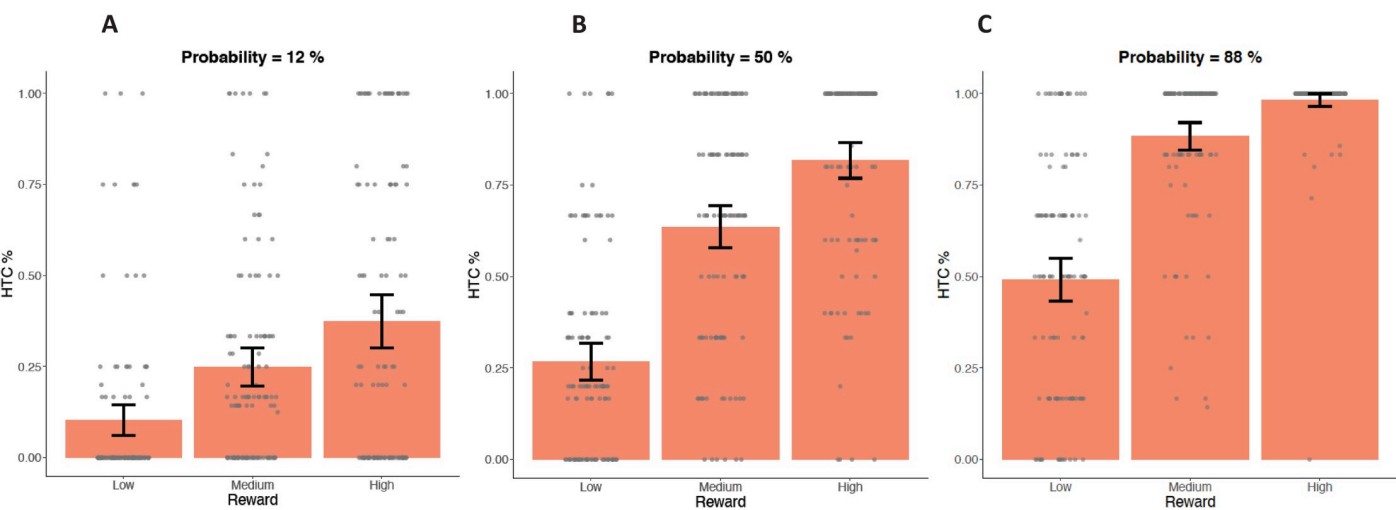

**Fig 3. Percentage of hard-task-choices (HTC) within the original EEfRT in Study 1.** Comparison of trials with low probability of reward attainment (left; A), medium probability of reward attainment (middle; B) and high probability of reward attainment (right, C) within the original EEfRT. For each probability category all three categories of reward magnitude (low / medium / high) are displayed. Data points are added as dots for individual scores. Error bars depict a 95% confidence interval (CI) of the mean.

of mean number of clicks for smaller reward magnitudes (see Fig 4). The factor MaxMot also reached significance (B = 0.61, $\chi^2(1) = 149.48$, $p < .001$), indicating that participants with greater motoric ability as measured within the motoric trials exerted more clicks within the modified EEfRT.

### 3.3 Experiment 2

**3.3.1 Original EEfRT—validity of basic task variables.** Participants on average chose the hard task in 53.31% of all trials (*SD* = 19.21%; Range = 0–100%). We computed a GEE model

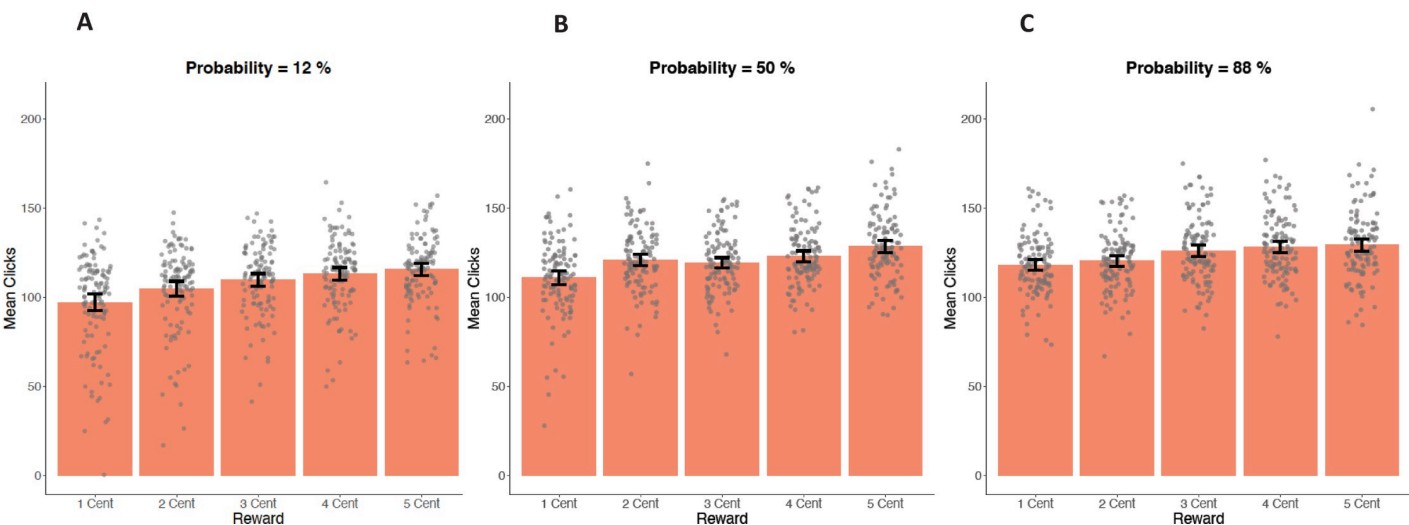

**Fig 4. Mean number of clicks within the modified EEfRT in Study 1.** Comparison of trials with low probability of reward attainment (left; **A**), medium probability of reward attainment (middle; **B**) and high probability of reward attainment (right; **C**) within the modified EEfRT. For each probability category all 5 different reward magnitudes (ranging from 1 cent (most left) to 5 cent (most right) are displayed. Data points are added as dots for individual scores. Error bars depict a 95% confidence interval (CI) of the mean.

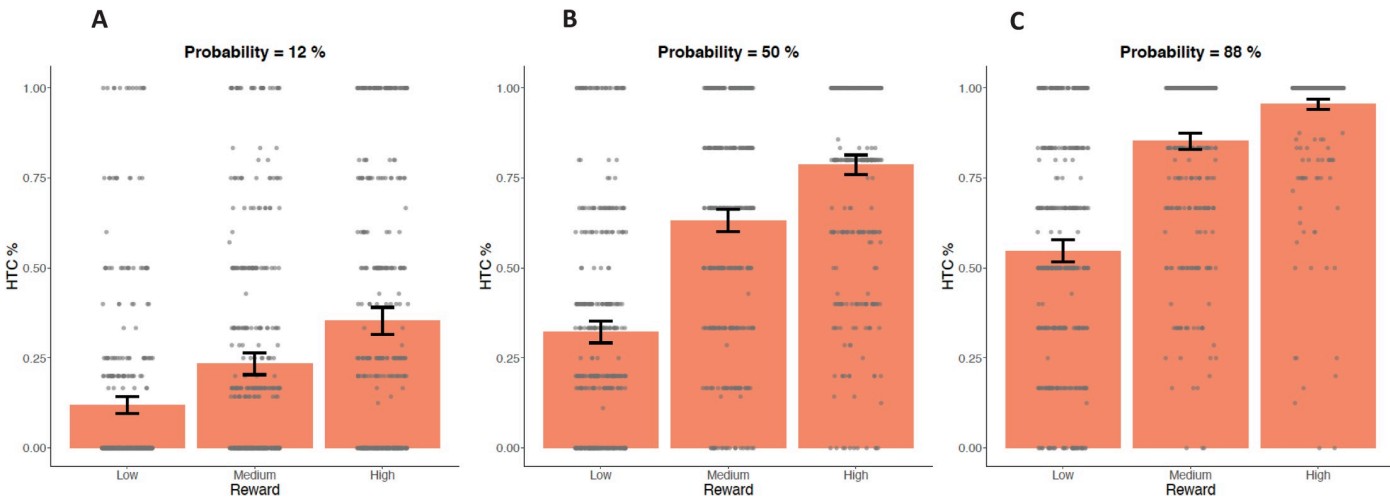

**Fig 5. Percentage of hard task choices (HTC) within the original EEfRT in Study 2.** Comparison of trials with low probability of reward attainment (left; A), medium probability of reward attainment (middle; B) and high probability of reward attainment (right; C) within the original EEfRT. For each probability category, all three reward magnitude categories (low / medium / high) are displayed. Data points are added as dots for individual scores. Error bars depict a 95% confidence interval (CI) of the mean.

to test the validity of basic task variables within the original EEfRT in Study 2 (see Table 3). The GEE Model examined main effects of task-dependent variables (reward magnitude, probability of reward attainment, and trial number) on the percentage of hard-task-choices. In line with previous studies and Study 1, significant positive main effects were found for reward magnitude and probability of reward attainment, and a significant negative main effect was found for trial number, indicating that all factors were predictors of the percentage of hard-task choices (all $p$s < .001). Furthermore, the interaction of reward and probability (often referred to as "expected value") reached significance ($p$ < .001), indicating that higher probability predicted a higher percentage of hard-task-choices with increasing reward magnitude (see Fig 5).

### 3.4 Personality correlations

To further validate the original and the modified EEfRT, we exploratorily correlated the percentage of hard-task-choices within the original EEfRT (Study 1 and Study 2) and the mean number of clicks within the modified EEfRT (Study 1) with personality traits as well as the mean number of pumps in the BART (Study 1). We compared all trial categories as well as difference scores (see Tables 4 and 5). Difference scores reflect individual differences in the degree to which participants' hard task choices or clicks are influenced by the probability of reward attainment and the reward magnitude, respectively. For instance, positive correlations between traits and difference scores would indicate that a higher trait value is associated with a stronger impact of probability of reward attainment or reward magnitude on task performance. Surprisingly, trait BAS did not correlate significantly with the percentage of hard-task-choices in trials with low probability of reward attainment within the original EEfRT in both studies, nor with the mean number of clicks within the modified EEfRT in trials with low probability of reward attainment (see Fig 6). However, in the original EEfRT in Study 1, trait BAS correlated negatively with the difference score between trials with high probability of reward attainment and low probability of reward attainment. Conversely, trait BIS correlated positively with both difference scores (high probability minus low / medium probability trials). Trait anticipatory pleasure correlated positively with number of hard task choices within trials with low probability of reward attainment, as well as negatively with the difference score

**Table 4. Correlations between the original EEfRT (percentage of hard-task -choices), the modified EEfRT (mean number of clicks) and trait variables (Study 1 and 2).**

| Trait variable | Reward Magnitude | | | | |
| --- | --- | --- | --- | --- | --- |
| **Original EEfRT (Study 1)** | low | medium | high | high-low | high-medium |
| BAS | .113 | .055 | -.055 | -.159 | -.147 |
| BIS | .066 | -.066 | -.135 | **-.180**[*] | -.086 |
| TEPS–anticipatory pleasure | **.235**[**] | .178 | .160 | -.097 | -.034 |
| TEPS–consummatory pleasure | .061 | .083 | .023 | -.041 | -.083 |
| UPPS-Urgency | -.131 | -.072 | -.021 | .112 | .070 |
| UPPS-Premeditation | -.067 | -.003 | .069 | .125 | .094 |
| UPPS-Perseverance | -.039 | -.042 | .090 | .115 | .175 |
| UPPS-Sensation Seeking | .117 | .107 | .074 | -.053 | -.049 |
| BART | .050 | .115 | .117 | .050 | -.004 |
| **Original EEfRT (Study 2)** | | | | | |
| BAS | -.026 | .006 | .020 | .046 | .016 |
| BIS | -.029 | -.035 | -.051 | -.013 | -.015 |
| UPPS-Urgency | .009 | -.009 | -.025 | -.032 | -.019 |
| UPPS-Premeditation | .017 | .016 | -.010 | -.027 | -.035 |
| UPPS-Perseverance | -.007 | .026 | -.010 | -.001 | -.049 |
| UPPS-Sensation Seeking | .036 | .079 | .092 | .043 | .005 |
| **Modified EEfRT (Study 1)** | | | | | |
| BAS | .046 | .006 | -.006 | -.089 | -.044 |
| BIS | .101 | .082 | .069 | -.060 | -.037 |
| TEPS–anticipatory pleasure | .080 | -.016 | -.020 | -.168 | -.014 |
| TEPS–consummatory pleasure | .069 | .042 | .062 | -.018 | .072 |
| UPPS-Urgency | -.017 | -.031 | -.054 | -.059 | -.082 |
| UPPS-Premeditation | -.060 | -.051 | -.041 | .037 | .032 |
| UPPS-Perseverance | -.150 | -.095 | -.060 | .159 | .115 |
| UPPS-Sensation Seeking | .024 | .035 | .055 | .048 | .071 |
| BART | .154 | **.224**[*] | **.253**[**] | .145 | .120 |

*Note.* EEfRT = Effort Expenditure for Rewards Task; BAS = Behavioral Activation System Scale; BIS = Behavioral Inhibition System Scale; TEPS: Temporal Experience of Pleasure Scale; UPPS = Urgency, Premeditation, Perseverance, and Sensation Seeking Impulsive Behavior Scale; BART = Balloon Analogue Risk Task. Correlations with (unadjusted) $p < .05$ are printed in bold. The significance level adjusted for multiple comparisons ($k = 5$ hypothesis tests for each correlate) was set to $p < .01$.

[*] $p < .05$

[**] $p < .01$

between trials with high probability of reward attainment and trials with low probability of reward attainment. Out of these correlations, only one was similar when comparing trials with different reward magnitudes. Trait anticipatory pleasure correlated positively with number of hard-task-choices within trials with low reward magnitude. Trait BIS correlated negatively with the difference score between trials with high reward magnitude and low reward magnitude Importantly, none of these correlations could be replicated in Study 2.

Note, however, that we did not administer the TEPS questionnaire and the BART in Study 2. The analyses for the modified EEfRT, which was administrated in Study 1 only, revealed a different pattern of results compared to the original EEfRT. Trait anticipatory pleasure correlated negatively with the difference score between trials with high probability of reward attainment and trials with medium probability of reward attainment. Moreover, risk-taking behavior as measured via the BART correlated positively with the mean number of clicks in trials with medium and high probabilities of reward attainment. This finding was similar when

**Table 5. Correlations between the original EEfRT (percentage of hard-task -choices), the modified EEfRT (mean number of clicks) and trait variables (Study 1 and 2).**

| Trait variable | Reward Probability | | | | |
|---|---|---|---|---|---|
| **Original EEfRT (Study 1)** | **12%** | **50%** | **88%** | **88–12%** | **88–50%** |
| BAS | .148 | .018 | -.073 | **-.181**$^{*}$ | -.074 |
| BIS | -.114 | -.105 | .165 | **.203**$^{*}$ | **.238**$^{**}$ |
| TEPS–anticipatory pleasure | **.291**$^{**}$ | .164 | .047 | **-.243**$^{**}$ | -.143 |
| TEPS–consummatory pleasure | .154 | -.007 | -.010 | -.149 | .000 |
| UPPS-Urgency | -.095 | -.066 | -.065 | .050 | .023 |
| UPPS-Premeditation | .058 | .016 | -.131 | -.131 | -.116 |
| UPPS-Perseverance | .029 | -.039 | -.006 | -.031 | .037 |
| UPPS-Sensation Seeking | .115 | .086 | .095 | -.051 | -.023 |
| BART | .021 | .123 | .148 | .068 | -.023 |
| **Original EEfRT (Study 2)** | | | | | |
| BAS | -.024 | -.022 | .059 | .060 | .079 |
| BIS | -.066 | -.027 | .004 | .063 | .036 |
| UPPS-Urgency | -.035 | -.015 | .050 | .064 | .061 |
| UPPS-Premeditation | .006 | .023 | -.017 | -.017 | -.043 |
| UPPS-Perseverance | -.018 | .051 | -.026 | .000 | -.084 |
| UPPS-Sensation Seeking | .025 | .065 | **.105**$^{*}$ | .044 | .015 |
| **Modified EEfRT (Study 1)** | | | | | |
| BAS | .015 | .045 | -.007 | -.024 | -.123 |
| BIS | .064 | .108 | .075 | .001 | -.071 |
| TEPS–anticipatory pleasure | .075 | .037 | -.055 | -.145 | **-.222**$^{*}$ |
| TEPS–consummatory pleasure | .049 | .080 | .048 | -.010 | -.072 |
| UPPS-Urgency | .016 | -.040 | -.081 | -.101 | -.103 |
| UPPS-Premeditation | -.032 | -.036 | -.080 | -.044 | -.111 |
| UPPS-Perseverance | -.064 | -.123 | -.119 | -.046 | .000 |
| UPPS-Sensation Seeking | .017 | .041 | .055 | .036 | .037 |
| BART | .126 | **.230**$^{*}$ | **.249**$^{**}$ | .106 | .067 |

*Note*. EEfRT = Effort Expenditure for Rewards Task; BAS = Behavioral Activation System Scale; BIS = Behavioral Inhibition System Scale; TEPS: Temporal Experience of Pleasure Scale; UPPS = Urgency, Premeditation, Perseverance, and Sensation Seeking Impulsive Behavior Scale; BART = Balloon Analogue Risk Task. Correlations with (unadjusted) $p < .05$ are printed in bold. The significance level adjusted for multiple comparisons ($k = 5$ hypothesis tests for each correlate) was set to $p < .01$.

$^{*} p < .05$

$^{**} p < .01$

examining reward magnitudes. Risk-taking behavior (BART) correlated positively with the mean number of clicks in trials with medium and high reward magnitudes. Across studies, tasks, and task parameters, the UPPS scales were largely unrelated to task performance.

## 3.5 Secondary analyses

As Study 1 is one of the first studies to test two versions of the EEfRT within one experimental design, we further evaluated the validity of both tasks by exploratorily correlating the main dependent variables of both task versions within all three different probability of reward attainment categories (low: 12% / medium: 50% / high: 88%; see Fig 7) and all three different reward magnitude categories (low / medium / high). The correlations between the matching dependent variables of both tasks were significant for all probabilities of reward attainment ($r$ ranging from .192 - .305). Furthermore, the correlation between the matching dependent

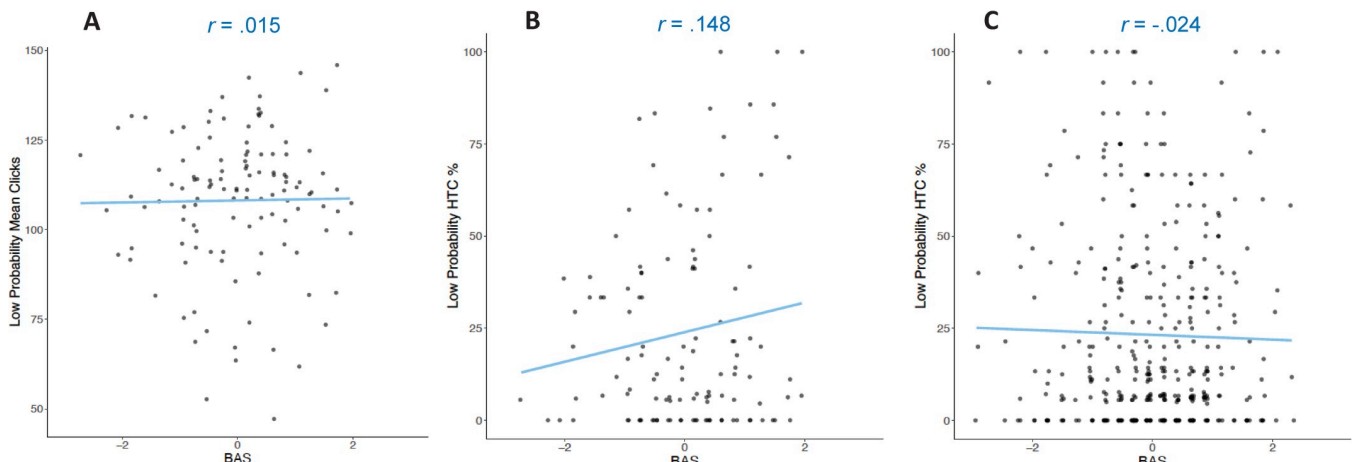

**Fig 6. Pearson correlations between trait BAS scores (z-standardized) and task performance (Study 1 and 2).** Correlations for the mean number of clicks within trials with low probability of reward attainment in the modified EEfRT (**A:** Study 1) and percentage of hard task choices within trials with low probability of reward attainment in the original EEfRT (**B:** Study 1; **C:** Study 2). All depicted correlations were nonsignificant with $p > .05$.

variables of both tasks was significant for trials with medium ($r = .277$) and high reward magnitudes ($r = .218$). Note that some discriminant correlations of non-matching reward probabilities or reward magnitudes exceeded the respective convergent correlations. The results indicate an overall linkage between performance on both task versions (albeit only small to moderate in size considering that the same construct should be measured). Subjects who choose the hard task more often on the original EEfRT also exert more clicks within the modified EEfRT (see Table 6).

Additionally, we asked participants to self-evaluate aspects that might have influenced their effort allocation individually for both task versions and asked them about their motivation to win money throughout the whole study. We then exploratorily correlated theses evaluations to

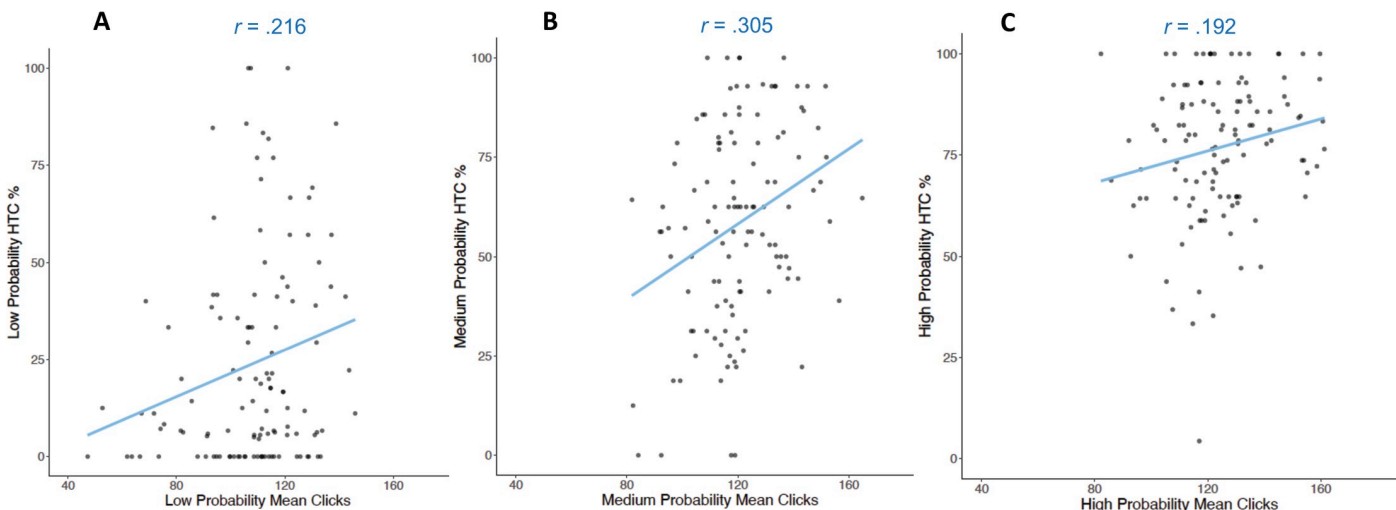

**Fig 7. Pearson correlations between task performance in both tasks in Study 1.** Correlations between the percentage of hard-task-choices (HTC %) within the original EEfRT and the mean number of clicks within the modified EEfRT for **A:** Trials with low probability of reward attainment (= 12%), **B:** Medium probability of reward attainment (= 50%) and **C:** High probability of reward attainment (= 88%). All depicted correlations were significant with $p < .05$.

**Table 6. Zero-order correlations between the original EEfRT (percentage of hard-task -choices) and the modified EEfRT (mean number of clicks) for different probabilities of reward attainment and different reward magnitudes.**

| | Reward Probability | | | | |
|---|---|---|---|---|---|
| **Task** | **Modified EEfRT** | | | | |
| **Original EEfRT** | 12% | 50% | 88% | 88–12% | 88–50% |
| 12% | **.216**\* | .085 | -.046 | **-.302**\*\* | **-.313**\*\* |
| 50% | **.340**\*\* | **.305**\*\* | **.221**\* | -.176 | **-.180**\* |
| 88% | .093 | .140 | **.192**\* | .086 | .138 |
| 88–12% | -.146 | .004 | .156 | **.331**\*\* | **.372**\*\* |
| 88–50% | **-.300**\*\* | **-.227**\* | -.097 | **.256**\*\* | **.299**\*\* |
| | **Reward Magnitude** | | | | |
| **Task** | **Modified EEfRT** | | | | |
| **Original EEfRT** | low | medium | high | high-low | high-medium |
| low | .160 | .043 | .020 | **-.242**\*\* | -.077 |
| medium | **.324**\*\* | **.277**\*\* | **.276**\*\* | -.108 | .022 |
| high | **.215**\* | **.210**\* | **.218**\* | -.015 | .047 |
| high-low | .024 | .137 | .166 | **.228**\* | .116 |
| high-medium | -.162 | -.104 | -.092 | .129 | .032 |

*Notes*. EEfRT = Effort Expenditure for Rewards Task. Correlations with (unadjusted) $p < .05$ are printed in bold. The adjusted significance level was set to $p < .01$.

\* p < .05

\*\* *p* < .01

**Table 7. Zero-order correlations between the original EEfRT (percentage of hard-task -choices), the modified EEfRT (mean number of clicks) and follow-up questions concerning task strategy and motivation in Study 1 for different probabilities of reward attainment.**

| Strategy and motivation | Reward Probability | | | | |
|---|---|---|---|---|---|
| **Original EEfRT** | **12%** | **50%** | **88%** | **88–12%** | **88–50%** |
| Reward | .118 | **.219**\* | .041 | -.086 | **-.207**\* |
| Probability | **-.513**\*\* | **-.232**\* | .128 | **.552**\*\* | **.348**\*\* |
| Fatigue | -.092 | -.082 | -.078 | .040 | .030 |
| Resting fingers | **-.199**\* | -.156 | -.093 | .130 | .100 |
| Motivation to win money | -.085 | -.002 | .069 | .120 | .054 |
| **Modified EEfRT** | | | | | |
| Reward | .140 | .172 | .155 | -.007 | -.026 |
| Probability | **-.200**\* | .017 | .144 | **.382**\*\* | **.312**\*\* |
| Fatigue | -.122 | -.055 | .030 | .175 | **.204**\* |
| Resting fingers | -.171 | -.140 | -.053 | .149 | **.201**\* |
| Motivation to win money | **.192**\* | **.231**\* | **.238**\*\* | .015 | .035 |

*Notes*. EEfRT = Effort Expenditure for Rewards Task. The following five-point Likert-scales ranging from "not at all" (1) to "a lot" (5) were administered separately for each task: Reward: "How much did the reward magnitudes influence you on the task?"; Probability: "How much did the probabilities of reward attainment influence you on the task?"; Fatigue: "How much did Fatigue influence you on the task?"; Resting Fingers: "How much did attempts to rest your fingers influence you on the task?". The last question (same response format) was asked only once (i.e., not separately for each task): Money: "How motivated were you to win money throughout the whole study?". Correlations with (unadjusted) $p < .05$ are printed in bold. The significance level adjusted for multiple comparisons ($k = 5$ hypothesis tests for each correlate) was set to $p < .01$.

\* *p* < .05

\*\* *p* < .01

**Table 8. Zero-order correlations between the original EEfRT (percentage of hard-task -choices), the modified EEfRT (mean number of clicks) and follow-up questions concerning task strategy and motivation in Study 1 for different reward magnitudes.**

| Strategy and motivation | Reward Magnitude | | | | |
|---|---|---|---|---|---|
| **Original EEfRT** | low | medium | high | high-low | high-medium |
| Reward | .029 | .153 | **.284**\*\* | **.212**\* | .162 |
| Probability | **-.204**\* | **-.243**\*\* | **-.389**\*\* | -.129 | -.177 |
| Fatigue | -.065 | -.115 | -.086 | -.009 | .045 |
| Resting fingers | -.108 | **-.231**\* | -.152 | -.022 | .117 |
| Motivation to win money | -.002 | -.024 | -.052 | -.043 | -.035 |
| **Modified EEfRT** | | | | | |
| Reward | .116 | **.190**\* | **.196**\* | .118 | .039 |
| Probability | -.123 | .020 | .059 | **.306**\*\* | .139 |
| Fatigue | -.100 | -.022 | -.023 | .134 | -.006 |
| Resting fingers | -.163 | -.124 | -.096 | .123 | .087 |
| Motivation to win money | .177 | **.253**\*\* | **.271**\*\* | .136 | .088 |

*Notes.* EEfRT = Effort Expenditure for Rewards Task. The following five-point Likert-scales ranging from "not at all" (1) to "a lot" (5) were administered separately for each task: Reward: "How much did the reward magnitudes influence you on the task?"; Probability: "How much did the probabilities of reward attainment influence you on the task?"; Fatigue: "How much did Fatigue influence you on the task?"; Resting Fingers: "How much did attempts to rest your fingers influence you on the task?". The last question (same response format) was asked only once (i.e., not separately for each task): Money: "How motivated were you to win money throughout the whole study?". Correlations with (unadjusted) $p < .05$ are printed in bold. The significance level adjusted for multiple comparisons ($k = 5$ hypothesis tests for each correlate) was set to $p < .01$.

\* $p < .05$

\*\* $p < .01$

their actual effort allocation in Study 1 comparing different trial categories and difference scores (see Tables 7 and 8). In line with our GEE analysis, which indicated probability of reward attainment to be strongly connected to actual task performance, participants'self-evaluated importance of this factor for their task performance correlated moderately to strongly with various trial categories in both task versions of the EEfRT (see Tables 7 and 8). The self-evaluated importance of reward magnitude was less strongly associated with performance in both task versions, although some moderately sized correlations emerged. When correlating participants'self-evaluated importance of fatigue for their task performance throughout the task, only one significant effect was observed. The number of clicks within the modified EEfRT correlated positively with the difference score between trials with high probability of reward attainment and medium probability of reward attainment. When correlating participants'self-evaluated importance of resting their fingers for their performance throughout the modified EEfRT, this evaluation also correlated significantly with the difference score between trials with high probability of reward attainment and trials with medium probability of reward attainment. A comparable result pattern was found for the original EEfRT. The strategy to rest their fingers was especially negatively related to hard-task-choices in trials with low reward probabilities and medium reward magnitudes. Finally, when correlating participants'motivation to win money throughout the whole study, the mean number of clicks within the modified EEfRT correlated significantly within trials with all probability levels, as well as with trials with medium and high reward magnitudes, indicating that participants with high motivation to win money performed more clicks in almost all trial categories (see Tables 7 and 8). In contrast to the modified EEfRT, the percentage of hard-task-choices within the original EEfRT did not correlate with participants'motivation to win money in any trial category.

## 4. Discussion

In the present study, we aimed to (1) validate the original EEfRT [21] and a modified version of the EEfRT [24] as measures of approach motivation by directly comparing both versions within one experimental design (Study 1) and to replicate the reliability and validity of the original EEfRT within a large sample (Study 2). We further aimed to (2) test the correlations between self-reported personality traits and behavioral measurements for different trial categories and difference scores, as well as between self-reported strategy usage and motivation and task performance in an exploratory fashion. We will now discuss the implication of the current findings.

### 4.1 Reliability and validity of the original and modified version of the EEfRT

Supporting the results of previous studies [14, 22, 24], both the original EEfRT and the modified EEfRT showed overall good split-half reliability, indicating that both versions are producing reliable results. In terms of validity, our results are mixed. The basic validity of both tasks as measures of reward-dependent approach motivation received further support from the GEE models in both studies as the basic task variables are in line with previous studies. In particular, we replicated the typical pattern of effects of reward magnitude, probability of reward attainment, and trial number on the mean number of clicks (modified EEfRT) and the percentage of hard-task-choices (original EEfRT). Furthermore, we found that the two versions of the EEfRT intercorrelated significantly within all three matching probability of reward attainment categories as well as in trials with medium and high reward magnitudes in Study 1. However, these correlations have to be considered small to moderate given that the same construct is supposed to be measured. Thus, differences between the two tasks need to be considered and will be discussed below.

Regarding the relationship between self-reported personality traits and behavioral task measures, our correlations in both studies showed only very weak support for such a link. Only in Study 1, trait BAS and trait anticipatory pleasure correlated significantly with the percentage of hard task choices within the original EEfRT and with the mean number of clicks within the modified EEfRT, for some task parameters. However, in contrast to our expectations, the correlating task parameters were not fully consistent with previous studies [23, 24]. Furthermore, trait BIS correlated with task performance in the original EEfRT in Study 1, indicating that trait BIS moderated the dependence of hard-task choices on reward magnitude (negatively related to BIS) and probability (positively related to BIS). However, as we analyzed the correlations of trait BAS and trait BIS for the original EEfRT within a larger sample in Study 2, none of these correlations replicated. Furthermore, other self-reported traits—especially impulsivity as measured via the UPPS as well as consummatory pleasure as measured via the TEPS—did not show meaningful correlations with the behavioral measures in either study.

Overall, these findings raise further questions about the existence and magnitude of such links, supporting studies not replicating them [22, 25, 26]. In particular, some correlations in previous work as well as Study 1 may have been overestimated due to random sampling error. Significant correlations may be at least partly attributable to the large number of possible correlations between task parameters and self-report measures. Supporting this notion, when correcting for multiple testing, only some effects of trait anticipatory pleasure and BIS remained significant, indicating that some of our findings might be false positives. Analyzing the correlations between performance on the BART and both task versions of the EEfRT in Study 1 revealed unexpected correlations between the mean number of clicks within the modified

EEfRT and the mean number of pumps within the BART, indicating that risk-taking behavior might have impacted task performance as well, although this requires further replication.

Taken together, the results of our data show a very mixed pattern regarding the validity of the EEfRT. Therefore, our results support a multiply determined view on behavioral measurements [42]. In line with this, previous studies indicate that effort allocation within the EEfRT can be manipulated by a wide range of factors, ranging from mood inductions [23] over neurophysiological manipulations [24, 28, 45] to the influence of reduced motivation [21], or the intake of caffeine [45]. So how does a person decide whether to increase effort to potentially gain a greater monetary reward within the EEfRT? Our mixed pattern of results shows that there is no simple answer to this question. Especially the impact of reward attributes hints at a complex pattern behind participants' decisions and at the importance of individual reward evaluation.

Reward-based decision making is not a uniform process, it can rather be described as a set of distinct cognitive processes, which together direct the evaluation of a reward and thus form a person's decisions within a concrete situation. According to Orsini et al. [61], reward-based (or "value-based") decision-making is comprised of three phases: 1. Decision representation (different options are identified, as are the costs and benefits associated with each option) and option valuation (each option is also assessed in terms of its subjective value in the moment of the decision), 2. action selection and 3. outcome evaluation (the value of the outcome of a choice is compared with the expected value of that outcome). It is reasonable to assume that the evaluation of potential benefits and costs can differ greatly between participants. Importantly, these individual differences may be insufficiently captured by typical personality trait questionnaires. A potentially important factor is the type of reward and how much a person values this reward. Real-life reward types include e.g., social [62], physical [63, 64], and recreational [65, 66] rewards and their valuation has been successfully differentiated via self-reports [67]. As stated above, Lopez-Gamundi & Wardle [51] were able to show that participants chose the hard task more often within a modified version of the EEfRT using cognitive tasks (C-EEfRT), although participants described the modified version as more difficult. The cognitive challenge of the modified version might have been rewarding in itself (although the monetary reward magnitude was unchanged). These results indicate that "costs" and "benefits" within a task can also be related to properties of the task itself.

To reach a better understanding of the self-evaluated aspects which might have influenced participants decisions, we asked them a series of questions about their strategies and motivation at the end of Study 1. We were able to show that effort allocation on both task versions was impacted by the self-evaluated importance of probability of reward attainment and reward magnitude, indicating that participants show some awareness of the factors that impact their behavior. Surprisingly, participants' self-evaluated motivation to win money throughout the whole study correlated positively only with the mean number of clicks within the modified EEfRT, in all three categories of probability of reward attainment as well as in trials with medium and high reward magnitudes. The percentage of hard-task-choices within the original EEfRT was not correlated with this self-evaluated monetary motivation. These results indicate that the individual evaluation of "costs" and "benefits" differs between both versions of the EEfRT, and hints at a potentially better validity of the modified EEfRT.

Lastly, one should also consider the nature of the questionnaires which assess personality traits, like trait BAS [6, 55], trait anticipatory pleasure [7] or trait impulsivity [56, 57]: These questionnaires consist of questions about various different situations, most of them complex real-life situations. Linking those scales to behavior in one artificial experimental situation might be rather difficult. There are several potential reasons for the lack of consistent associations. These pertain to (1) the validity of the questionnaire measures (e.g., self-reported traits

only reflect one aspect of personality), (2) the validity of the tasks (e.g., factors other than approach motivation may substantially affect task performance), and (3) the similarity of the measured constructs. Regarding the last point, one issue that deserves particular attention is the breadth of global trait measures as compared to the narrowness of task behavior in one laboratory situation. For instance, the EEfRT could be a valid measure of approach motivation in this circumscribed situation and nevertheless be too narrow for the assessment of broad trait-like behavioral tendencies.

## 4.2 Limitations and future directions

Although we analyzed two rather large samples to test the reliability and validity of the original and the modified version of the EEfRT and our study is one of the first to directly compare two versions of the EEfRT within one experimental design, there are still some limitations to our study.

First, although we tried to stick as close to the original version of the EEfRT as possible [21], there is still a noteworthy adaption, which might have impacted participants'behavior in our "original" EEfRT strongly. The adaption is based on a study by Hughes et al. [46], who decided to pay participants a percentage of the virtually won money instead of paying participants the money which they have won on two random trials [21]. We followed this adaption, as we expected the non-random payment to increase participants'overall approach motivation. However, we did not expect this adaption to change the basic response pattern in any significant way, which is also supported by our results replicating the basic predictors (i.e., reward attributes). Nonetheless, as we stated in the introduction, many adaptions of the EEfRT have been used in various studies, ranging from reduced complexity by fixing the monetary rewards [37], or by removing trials with low probability of reward attainment [38] to the addition of "loose"–trials [52], or the addition of a social component [47]. Thus, we cannot rule out that our modification might have caused a significant change in behavior within the original EEfRT. In this regard, our modification might have impacted participants' strategic behavior, as the random payment introduced by the original EEfRT [21] might reduce strategic task choices compared to our version of the EEfRT. Therefore, another direct comparison of these two task versions is needed in future studies.

Second, although both task versions used in Study 1 correlated significantly, suggesting some overlap, the correlations ranging from $r = .160$ to $.305$ between matching variables indicate that the variables captured by the two tasks also differ substantially. Fig 7 indicates that this might partly be a result of floor-, and ceiling-effects regarding the effort allocation within original EEfRT (especially in trials with low and high probability of reward attainment). However, our findings suggest further substantial differences between both task versions. First, motoric abilities are more strongly related to the number of clicks in the modified EEfRT compared to hard task choices in the original EEfRT (see Table 3). This may indicate that the task versions differ in terms of physical demands. Therefore, future studies using the modified EEfRT should always include a measure of motoric abilities. Second, in the original EEfRT, the effects of reward magnitude on hard task choices are strongest for higher probabilities of reward attainment. In contrast, in the modified EEfRT, effects of reward magnitude on the number of clicks are largest for smaller probabilities of reward attainment (see Table 3). Thus, response behavior has been altered by the modifications made to the original task. Future studies should always consider that any change made to the original EEfRT e.g., to make the task fit to the experimental setting could lead to substantial changes in behavior and should therefore carefully compare any new version of the EEfRT to the original version. The aforementioned floor- and ceiling-effects could be a general problem of the original EEfRT–at least

when testing young healthy participants. These effects should also be considered, for instance, when comparing healthy participants to patients with impaired approach motivation.

Third, although we were able to test the original EEfRT within a large sample in Study 2, the comparison of both studies is limited. As stated in the methods section, the original EEfRT was part of a larger genetic study in Study 2 and task administration took place approximately 4 hours after start of the testing session. It is reasonable to assume that completing the task after 4 hours of testing might have influenced participants' fatigue or boredom. Study 1 on the other hand lasted for only one hour, which might have led to less fatigue or boredom.

Fourth, both samples consisted mainly of young healthy students and both samples were not balanced in terms of age or gender. Thus, it remains an open question whether the results of our research can be generalized to broader populations. As the original EEfRT has been demonstrated to be a sensitive tool for detecting reduced approach motivation within various clinical samples [32–38], we cannot rule out that our homogeneous sample reduced the range of responses and the results we found regarding the reliability and validity of the EEfRT could show a different pattern for other populations, e.g., within clinical samples.

Fifth, as we discussed above, the rather mixed results regarding the validity of the original and the modified EEfRT might be a result of correlating scales from self-reports summarizing a large set of real-life situations and behavioral measurements within one laboratory task. The EEfRT assesses the investment of physical effort for monetary rewards. Other forms of effort as well as other forms of rewards should be introduced and tested. Furthermore, we suggest that assessing approach motivation should not be limited to one specific "cost" and one specific "benefit". For instance, future work might use a range of tasks or a range of variations of the EEfRT and calculate scales comparable to questionnaires assessing personality traits. The EEfRT offers a solid foundation to probe other forms of "costs" (e.g., the C-EEfRT which uses cognitive costs) [51] and a variety of different "benefits" (e.g., food portions) [50].

## 4.3 Conclusion

Taken together, our findings provide additional support for the split-half reliability of the original and the modified version of the EEfRT. Furthermore, the correlations between both task versions provide evidence for some overlap of the two tasks. However, these correlations have to be considered small to moderate given that the same construct was targeted. Thus, these findings also hint at substantial differences between the two tasks. The results regarding the validity of the tasks are mixed. While the basic predictors of both task versions replicated well in both studies and are also supported by participants' self-evaluated importance of these factors, we were not able to replicate previous findings linking trait BAS and trait anticipatory pleasure to effort allocation within both versions of the EEfRT as only some performance parameters in Study 1 correlated with self-reported personality traits. Study 2 with a larger sample did not reveal any correlation involving trait BAS for the original EEfRT. Furthermore, self-evaluated motivation hints at a possible advantage of the modified EEfRT regarding its validity. Our results indicate a complex interplay of personality traits, task properties, and reward attributes. Furthermore, they highlight the importance of analyzing the reliability and validity of the EEfRT and any modification applied to the task.

## Acknowledgments

This research did not receive any specific grant from funding agencies in the public, commercial, or not-for-profit sectors. There were no conflicts of interest. The studies and the corresponding hypotheses were not preregistered. Datasets and syntax will be made publicly

available here: https://osf.io/35k2w/. Furthermore, we would like to thank Karoline Rosenkranz for supporting our project by collecting the data of study 2.

## Author Contributions

**Conceptualization:** Hanno Andreas Ohmann, Niclas Kuper.

**Data curation:** Hanno Andreas Ohmann.

**Formal analysis:** Hanno Andreas Ohmann, Niclas Kuper.

**Investigation:** Hanno Andreas Ohmann.

**Methodology:** Hanno Andreas Ohmann.

**Project administration:** Hanno Andreas Ohmann, Jan Wacker.

**Supervision:** Jan Wacker.

**Visualization:** Niclas Kuper.

**Writing – original draft:** Hanno Andreas Ohmann.

**Writing – review & editing:** Hanno Andreas Ohmann, Niclas Kuper, Jan Wacker.

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
