## [Decision Letter · Decision Letter 0]

2 Jun 2021

PONE-D-21-08512

Examining the reliability and validity of two versions of the Effort-Expenditure for Rewards Task (EEfRT)

PLOS ONE

Dear Dr. Ohmann,

Thank you for submitting your manuscript to PLOS ONE. After careful consideration, we feel that it has merit but does not fully meet PLOS ONE’s publication criteria as it currently stands. Therefore, we invite you to submit a revised version of the manuscript that addresses the points raised during the review process.

Both reviewers consider this study positively and useful for anyone using the EEfRT. They also suggest a variety of minor changes to improve the quality of the manuscript. I agree with both of them. Thus I suggest the authors revise their original manuscript and resubmit their revision with the detailed changes to the journal.

We look forward to receiving your revised manuscript.

Kind regards,

Vilfredo De Pascalis

Academic Editor

PLOS ONE

Journal Requirements:

3. Thank you for including your ethics statement:  "The study has been approved by the local Ethics Committee in written form.".   

4. Please improve statistical reporting and replace commas with decimals when reporting statistics, e.g., 78.3% instead of 78,3%. Our statistical reporting guidelines are available at https://journals.plos.org/plosone/s/submission-guidelines#loc-statistical-reporting.

Additional Editor Comments:

Both reviewers consider this study positively and useful for anyone using the EEfRT. They also suggest a variety of minor changes to improve the quality of the manuscript. I agree with both of them. Thus I suggest the authors revise their original manuscript and resubmit their revision with the detailed changes to the journal.

Reviewers' comments:

Reviewer's Responses to Questions

**Comments to the Author**

1. Is the manuscript technically sound, and do the data support the conclusions?

Reviewer #1: Partly

Reviewer #2: Yes

2. Has the statistical analysis been performed appropriately and rigorously? 

Reviewer #1: N/A

Reviewer #2: Yes

3. Have the authors made all data underlying the findings in their manuscript fully available?

Reviewer #1: Yes

Reviewer #2: Yes

4. Is the manuscript presented in an intelligible fashion and written in standard English?

Reviewer #1: Yes

Reviewer #2: Yes

5. Review Comments to the Author

Reviewer #1: This study represents an important, relatively large scale study of the psychometrics and convergent and discriminant validity of alternate forms of the EEfRT, a task that is often modified with surprisingly little examination of the effects of those modifications. This report is an important one for anyone using the EEfRT to consider, and I offer a variety of suggestions to strengthen its contribution to the literature.

INTRODUCTION

In the Hypotheses, the authors note that the reliability of the EEfRT appears promising from three previous studies; however, it would be helpful to note that based on their table, this is test-retest reliability specifically.

In the paragraph spanning pp. 11-12, what were the actual correlations in the cited studies between EEfRT behavior and self-report measures? Having those relayed in the text would allow the reader to gauge immediately whether the authors’ assertions about the expected effect size were appropriate. In general, it would be appropriate to expect some shrinkage from published effect sizes, especially with a larger sample (which then has a smaller critical r value for a particular alpha level). Furthermore, Study 2’s Participants description seems to assume a smaller critical correlation without it being introduced here.

METHOD (with references largely confined to Study 1)

To what extent do the rather rigorous exclusion criteria match those used in other studies with healthy samples?

Would changing the distribution of money from a randomly sampled two trials to 5% of the total winnings on the EEfRT constitute a modification to the overall procedure?

Nice job of using randomized task order to prevent order effects from influencing behavior. Why was the BART always in the middle of the task order?

Though the authors cite a couple of studies for the BART’s automatic response procedure, it would seem that version of the task would compromise the motoric response component of the BART that might make it most comparable to the EEfRT. To what extent is that modified version of the BART consistent with the motor-intensive original version that requires key presses for each pump, both in terms of scores and correlations with external variables in theoretically expected ways?

In the paragraph describing the modified EEfRT, I now see how the awarding of 5% of the total money earned might constitute a substantial modification compared to awarding the money won on two randomly selected trials. When money is awarded as a function of the proportion of (hard) trials completed, strategic concerns are minimized (which was an integral part of Zald and Treadway’s reasoning for designing the task that way; see the middle of the second to last paragraph in the section describing the EEfRT in Treadway et al., 2009). However, when a flat 5% of the winnings are awarded, strategy plays a substantial role in what kinds of trials a person might choose to play. This discrepancy in extra compensation paradigms deserves explication.

Just to clarify, the maximum number of button presses across each of the 10 motoric ability trials was used as the MaxMot variable? Was this variable computed separately for each of the two iterations of this task?

What response function was used in the GEEs? The analyses of hard task choices should use some kind of binomial-based response rather than one assuming a linear response.

The GEEs should not use an independent working correlation matrix; that will assume there are no correlations among cells, which is incorrect in a design featuring within-subjects factors. Treadway et al. (2009) used an unstructured correlation matrix to fully account for the vagaries of the relationships among cells, but one can argue that correlation matrix estimates too many parameters to be stable. An autoregressive or m-dependent matrix might be plausible, as might an exchangeable matrix if the authors really wanted to cut down on the parameters estimated. However, the dependence among observations should be accounted for in the working correlation matrix to ensure reasonable standard errors. The marginal means are consistent even if a working correlations is misspecified, but the standard errors are not (e.g., Hin & Wang, 2009). However, the overall standard errors are not so extreme as to vitiate finding meaningful results in this study (per Table 3).

Did the authors ensure equal distributions of low, medium, and high reward magnitudes across original EEfRT trials based on their binnings presented in the Data Analysis section?

RESULTS

If the split half reliability was calculated for the EEfRT’s reliability, Cronbach’s alpha (which is the average of all possible split halves) could also be calculated unless the halves were carefully balanced in some way. However, this is a relatively minor point.

How did the authors adjust raw EEfRT scores in Study 1 for motoric activity? Did they analyze some sort of residualized task performance score?

If I recall correctly, the β column in Table 3 should actually be a B column, as those coefficients are unstandardized.

Figures 3-5 were really nicely done as a way to examine each participant’s data along with the measures of central tendency and variability.

In Table 4, what adjustment was made for families of multiple comparisons? Some adjustment needs to be made given the exploratory nature of these analyses, which would likely alter the presentation of the results on p. 27. To maintain the presentation, it might be reasonable to use one tabular asterisk for an uncorrected p < .05 and two tabular asterisks for a corrected p < .05.

In the first paragraph of the Secondary Analyses section, I would argue that these correlations are even moderate given that they ostensibly represent alternate versions of the same task. I also did not see the values of the correlations for the various reward magnitudes, only the reward probabilities reported in Figure 7 (which is a lovely visualization of the relevant data).

Tables 6 and 7 need corrections for multiple comparisons as well.

DISCUSSION

I believe the authors oversell substantially the comparability of the two versions of the EEfRT. Considerations of effect size are paramount in the first paragraph (e.g., difference between each form’s internal consistency and alternate forms’ correlations).

The discussion of self-report correlations may be a discussion of false positive errors given the lack of replicability of most of them across Studies 1 and 2 and the lack of corrections for multiple comparisons. The authors nod to this near the end of the second paragraph of the Discussion’s first section, but they can check how well their findings would survive correction for multiple comparisons.

I can appreciate how the authors contextualize the size of their adaptation of extra compensation allocation compared to other EEfRT adaptations. However, I think they may underestimate how that changes the (lack of) strategic considerations in the original EEfRT, which may explain some of the low correlations across EEfRT forms.

Good note that floor and ceiling effects in the original EEfRT might limit the size of the validity coefficients with the modified EEfRT.

MINOR POINTS

Many paragraphs span multiple pages. These must be broken so that they occupy ½ to ¾ of a single manuscript page so they contain a single idea and its elaborations.

REFERENCES FOR THIS REVIEW

Hin, L.-Y., & Wang, Y.-G. (2009). Working-correlation-structure identification in generalized estimating equations. Statistics in Medicine, 28, 642-658. https://doi.org/10.1002/sim.3489

Reviewer #2: Generally a well done paper. It is a useful set of results about a fairly widely used measure. It is also quite informative to show that two versions of what is ostensibly the same measure, may actually have fairly modest correlations.

In the discussion, I think the authors should explicitly mention all the personality traits that are not related to EEfRT, such as the UPPS. Although they do have some discussion of what it means that the behavioral measure is not really related to some important reward related measures of depression, I also think that they should provide more discussion of what it means that this behavioral measure is only minimally correlated with personality variables that have been shown to be related to depression, which the EEfRT is claimed to help measure.

I have a number of minor comments.

As they note, the measure was developed for use in distinguishing clinical samples from normals, so it might be problematic to only collect data on a young, healthy sample. They do note at one point, the problem they had with ceiling and floor effects. This might be a general problem with the EEfRT.

They aren't always as explicit as they should be in stating that some of their measures, such as the TEPS and the UPPS are essentially not correlated with the EEfRT in a normal sample.

Line 49. Culminated would be a better word choice than accumulated.

Line 75. Specifically would be a better word choice than Precisely.

Line 86. Say that one would expect relatively specific associations, but then don't give any idea of what those are.

Line 108. "During task performance", not "while task performance"

Line 279. "Then, after a ready – screen (1s, C) main screen for the trial showing a…" There is punctuation or words missing. Doesn’t' read properly.

Line 324 starting with "Furthermore, we…" Something is wrong with this sentence.

Line 387. "criterion" not "criterium"

Line 550. A correlation with a difference score is hard to interpret. Would be nice to explicitly state what it means.

Line 560. So basically almost none of the personality variables are correlated with either version of the EEfRT in a normal sample?

6. PLOS authors have the option to publish the peer review history of their article (what does this mean?). If published, this will include your full peer review and any attached files.

Reviewer #1: **Yes: **Stephen D. Benning

Reviewer #2: No

---

## [Author Response · Author response to Decision Letter 0]

19 Aug 2021

Response to Reviewer 1:

Reviewer #1: This study represents an important, relatively large scale study of the psychometrics and convergent and discriminant validity of alternate forms of the EEfRT, a task that is often modified with surprisingly little examination of the effects of those modifications. This report is an important one for anyone using the EEfRT to consider, and I offer a variety of suggestions to strengthen its contribution to the literature.

Thank You! Please note that line numbers in our response refer to the new “Manuscript” file (as line numbering is messed up within the tracked changes document.

INTRODUCTION

In the Hypotheses, the authors note that the reliability of the EEfRT appears promising from three previous studies; however, it would be helpful to note that based on their table, this is test-retest reliability specifically.

Thank you for raising this important issue. We clarified the types of reliability measured in previous studies as mentioned in the hypothesis (p. 11, l. 176-179).

In the paragraph spanning pp. 11-12, what were the actual correlations in the cited studies between EEfRT behavior and self-report measures? Having those relayed in the text would allow the reader to gauge immediately whether the authors’ assertions about the expected effect size were appropriate. In general, it would be appropriate to expect some shrinkage from published effect sizes, especially with a larger sample (which then has a smaller critical r value for a particular alpha level). Furthermore, Study 2’s Participants description seems to assume a smaller critical correlation without it being introduced here.

We fully agree and added the actual correlations to that paragraph (p. 11-12, l. 193-198). Indeed, we also expected some shrinkage from the published effect sizes. The smaller critical correlation expected in study 2 is partly based on the discrepancy found between published correlations and the correlations found in our own lab. We clarified this issue in the methods section of study 2 (p. 19-20, l. 402-406).

METHOD (with references largely confined to Study 1)

To what extent do the rather rigorous exclusion criteria match those used in other studies with healthy samples?

Our exclusion criteria within study 1 are in line with previous studies conducted within our lab (i.e., Ohmann et al., 2018; Ohmann et al., 2020) and we believe those criteria are commonly used in the literature. The stricter criteria within study 2 are due to the clinical setting and the various molecular genetic hypotheses which are not part of the reported project.

Would changing the distribution of money from a randomly sampled two trials to 5% of the total winnings on the EEfRT constitute a modification to the overall procedure?

See below.

Nice job of using randomized task order to prevent order effects from influencing behavior. Why was the BART always in the middle of the task order?

Both versions of the EEfRT demand rather high motor-intense reactions. Thus, the BART (without any motor-intense reaction) has been used in the middle to avoid motoric fatigue of the subjects throughout the whole study. We adjusted the methods section to point out this issue (p. 14, l. 258-260).

Though the authors cite a couple of studies for the BART’s automatic response procedure, it would seem that version of the task would compromise the motoric response component of the BART that might make it most comparable to the EEfRT. To what extent is that modified version of the BART consistent with the motor-intensive original version that requires key presses for each pump, both in terms of scores and correlations with external variables in theoretically expected ways?

The reactions to the BART should be seen as a different kind of reaction compared to the EEfRT, as the “pumps” of the original BART are not made under time pressure and each reaction could lead to the loss of the earned money within the specific trial (Lejuez et al., 2002). Thus, the task is not intended to measure physical effort. Instead, the BART (especially the automatic response procedure we used) measures impulsivity and the willingness of taking risks – which is the reason why we used this task. In this regard the automatic response version appears to be related to other measures in comparable fashion as the original version (for a comparison of both versions see Pleskac et al., 2008). 

In the paragraph describing the modified EEfRT, I now see how the awarding of 5% of the total money earned might constitute a substantial modification compared to awarding the money won on two randomly selected trials. When money is awarded as a function of the proportion of (hard) trials completed, strategic concerns are minimized (which was an integral part of Zald and Treadway’s reasoning for designing the task that way; see the middle of the second to last paragraph in the section describing the EEfRT in Treadway et al., 2009). However, when a flat 5% of the winnings are awarded, strategy plays a substantial role in what kinds of trials a person might choose to play. This discrepancy in extra compensation paradigms deserves explication.

Thank you for pointing out this important matter! We added a more critical review on this issue in the discussion (p. 41, l. 832-848).

Just to clarify, the maximum number of button presses across each of the 10 motoric ability trials was used as the MaxMot variable? Was this variable computed separately for each of the two iterations of this task?

Yes, the maximum number of button presses across each of the 10 motoric ability trials was used as the MaxMot variable and it was computed separately for each iteration of this task. As both versions of the EEfRT demand motor-intense reactions, we couldn`t rule out any kind of motoric fatigue. Therefore, we added two iterations of the motoric ability task (one before each version of the EEfRT). However, the maximum number of button presses did not differ significantly between both iterations.

What response function was used in the GEEs? The analyses of hard task choices should use some kind of binomial-based response rather than one assuming a linear response.

Yes, we used a binomial-based response for the hard task choices. We added this information to the methods section (p. 18, l. 373-375).

The GEEs should not use an independent working correlation matrix; that will assume there are no correlations among cells, which is incorrect in a design featuring within-subjects factors. Treadway et al. (2009) used an unstructured correlation matrix to fully account for the vagaries of the relationships among cells, but one can argue that correlation matrix estimates too many parameters to be stable. An autoregressive or m-dependent matrix might be plausible, as might an exchangeable matrix if the authors really wanted to cut down on the parameters estimated. However, the dependence among observations should be accounted for in the working correlation matrix to ensure reasonable standard errors. The marginal means are consistent even if a working correlations is misspecified, but the standard errors are not (e.g., Hin & Wang, 2009). However, the overall standard errors are not so extreme as to vitiate finding meaningful results in this study (per Table 3).

GEEs provide unbiased estimates and appropriate robust standard errors even in the case that the working correlation matrix is misspecified. We emphasize this in the text (p. 18, l. 368-372). In line with your suggestions, we now report results using an exchangeable correlation matrix (Table 3). Results using this correlation matrix were highly similar to our original results.

Did the authors ensure equal distributions of low, medium, and high reward magnitudes across original EEfRT trials based on their binnings presented in the Data Analysis section?

The distribution of low, medium and high reward magnitudes is equal across the binnings of the original EEfRT trials. 

RESULTS

If the split half reliability was calculated for the EEfRT’s reliability, Cronbach’s alpha (which is the average of all possible split halves) could also be calculated unless the halves were carefully balanced in some way. However, this is a relatively minor point.

Thank you for this suggestion. However, we intentionally chose the implemented split-half correlation over a measure of internal consistency. In particular, reliability was calculated by splitting by trial number (original task, with varying trial numbers per person) and block (modified task), yielding balanced test halves. We adjusted the method section to increase precision (p. 18, l. 363-365). Furthermore, we adjusted the report of reliabilities to show all relevant variables (Table 2)

How did the authors adjust raw EEfRT scores in Study 1 for motoric activity? Did they analyze some sort of residualized task performance score?

Yes, we residualized the task performance of the EEfRT (p. 21, l. 450-454). Within the GEEs we added the motoric abilities as an additional factor (MaxMot) to directly access its impact on task performance.

If I recall correctly, the β column in Table 3 should actually be a B column, as those coefficients are unstandardized.

Thank you for making us aware of this, we replaced all betas with Bs accordingly.

Figures 3-5 were really nicely done as a way to examine each participant’s data along with the measures of central tendency and variability.

Thank You!

In Table 4, what adjustment was made for families of multiple comparisons? Some adjustment needs to be made given the exploratory nature of these analyses, which would likely alter the presentation of the results on p. 27. To maintain the presentation, it might be reasonable to use one tabular asterisk for an uncorrected p < .05 and two tabular asterisks for a corrected p < .05.

Thank you for this suggestion. We implemented the suggested approach in Tables 4, 5, 6, 7, and 8. In particular, we Bonferroni-corrected the alpha level for each correlate (trait or strategy / motivation) separately (5 correlates each), yielding an adjusted alpha level of .01. Accordingly, we note in the respective table notes that * indicates p <.05 at the uncorrected significance level and ** indicates p <.01, corresponding to p <.05 at an alpha level corrected for multiple comparisons for each correlate.

In the first paragraph of the Secondary Analyses section, I would argue that these correlations are even moderate given that they ostensibly represent alternate versions of the same task. I also did not see the values of the correlations for the various reward magnitudes, only the reward probabilities reported in Figure 7 (which is a lovely visualization of the relevant data).

We agree and added an additional table (Table 6) containing all relevant correlations between task parameters of the two tasks.

Tables 6 and 7 need corrections for multiple comparisons as well.

Thank you for making us aware of this, we implemented this correction as described above. 

DISCUSSION

I believe the authors oversell substantially the comparability of the two versions of the EEfRT. Considerations of effect size are paramount in the first paragraph (e.g., difference between each form’s internal consistency and alternate forms’ correlations).

We adjusted the discussion and added a more critical view on this issue (p. 37, l. 741-746).

The discussion of self-report correlations may be a discussion of false positive errors given the lack of replicability of most of them across Studies 1 and 2 and the lack of corrections for multiple comparisons. The authors nod to this near the end of the second paragraph of the Discussion’s first section, but they can check how well their findings would survive correction for multiple comparisons.

We adjusted the discussion and added a more critical view on this issue based on the correction for multiple testing (p. 38, l. 761-767).

I can appreciate how the authors contextualize the size of their adaptation of extra compensation allocation compared to other EEfRT adaptations. However, I think they may underestimate how that changes the (lack of) strategic considerations in the original EEfRT, which may explain some of the low correlations across EEfRT forms.

We adjusted the discussion and added a more critical view on this issue (p. 41, l. 832-848).

Good note that floor and ceiling effects in the original EEfRT might limit the size of the validity coefficients with the modified EEfRT.

Thank You!

MINOR POINTS

Many paragraphs span multiple pages. These must be broken so that they occupy ½ to ¾ of a single manuscript page so they contain a single idea and its elaborations.

We tried to break paragraphs wherever possible and hope that the Ms is now easier to read.

REFERENCES FOR THIS REVIEW

Hin, L.-Y., & Wang, Y.-G. (2009). Working-correlation-structure identification in generalized estimating equations. Statistics in Medicine, 28, 642-658. https://doi.org/10.1002/sim.3489

References for this response

Lejuez CW, Richards JB, Read JP, Kahler CW, Ramsey SE, Stuart GL, et al. Evaluation of a behavioral measure of risk taking: The balloon analogue risk task (BART). J Exp Psychol Appl. 2002; 8(2): 75–84. doi: 10.1037/1076-898X.8.2.75

Ohmann HA, Kuper N, Wacker J. Left frontal anodal tDCS increases approach motivation depending on reward attributes. Neuropsychologia. 2018; 119: 417–423. doi: 10.1016/j.neuropsychologia.2018.09.002

Ohmann HA, Kuper N, Wacker J. A low dosage of the dopamine D2-receptor antagonist sulpiride affects effort allocation for reward regardless of trait extraversion. Personal Neurosci. 2020; 3: E7. doi: 10.1017/pen.2020.7

Pleskac TJ, Wallsten TS, Wang P, Lejuez CW. Development of an automatic response mode to improve the clinical utility of sequential risk-taking tasks. Exp Clin Psychopharmacol. 2008; 16(6): 555–564. doi: 10.1037/a0014245

Response to Reviewer 2:

Reviewer #2: Generally a well done paper. It is a useful set of results about a fairly widely used measure. It is also quite informative to show that two versions of what is ostensibly the same measure, may actually have fairly modest correlations.

Thank you! Please note that line numbers in our response refer to the new “Manuscript” file (as line numbering is messed up within the tracked changes document.

In the discussion, I think the authors should explicitly mention all the personality traits that are not related to EEfRT, such as the UPPS. Although they do have some discussion of what it means that the behavioral measure is not really related to some important reward related measures of depression, I also think that they should provide more discussion of what it means that this behavioral measure is only minimally correlated with personality variables that have been shown to be related to depression, which the EEfRT is claimed to help measure.

We agree and now explicitly mention these traits in the discussion (p. 38, l. 755-760). Furthermore, we expanded the discussion of the minimal correlations between behavioral measures and personality variables: We think that these correlations are still too exploratory in nature to be interpreted in terms of measurement of depression (via the EEfRT) and there are too many other possible factors impacting these correlations (p. 40, l. 814-826).

I have a number of minor comments.

As they note, the measure was developed for use in distinguishing clinical samples from normals, so it might be problematic to only collect data on a young, healthy sample. They do note at one point, the problem they had with ceiling and floor effects. This might be a general problem with the EEfRT.

Thank you for mentioning this issue. We adjusted the discussion (p. 42, l. 857-860).

They aren't always as explicit as they should be in stating that some of their measures, such as the TEPS and the UPPS are essentially not correlated with the EEfRT in a normal sample.

We adjusted the discussion as mentioned above (p. 38, l. 755-760).

Line 49. Culminated would be a better word choice than accumulated.

We corrected the term, thank you.

Line 75. Specifically would be a better word choice than Precisely.

We corrected the term, thank you.

Line 86. Say that one would expect relatively specific associations, but then don't give any idea of what those are.

We added an explanation to this section (p. 4-5, l. 89-92).

Line 108. "During task performance", not "while task performance"

We corrected the term, thank you.

Line 279. "Then, after a ready – screen (1s, C) main screen for the trial showing a…" There is punctuation or words missing. Doesn’t' read properly.

We increased the precision of the task description (p. 15, l. 289-291).

Line 324 starting with "Furthermore, we…" Something is wrong with this sentence.

We corrected that sentence (p. 17, l. 334).

Line 387. "criterion" not "criterium"

We corrected the term, thank you.

Line 550. A correlation with a difference score is hard to interpret. Would be nice to explicitly state what it means.

We added an explanation to that section (p. 28 l. 576-579).

Line 560. So basically almost none of the personality variables are correlated with either version of the EEfRT in a normal sample?

That’s correct. Overall, we increased precision of our manuscript regarding this issue according to your valuable input!

---

## [Decision Letter · Decision Letter 1]

29 Nov 2021

PONE-D-21-08512R1Examining the reliability and validity of two versions of the Effort-Expenditure for Rewards Task (EEfRT)PLOS ONE

Dear Dr. Ohmann,

Thank you for submitting your manuscript to PLOS ONE. After careful consideration, we feel that it has merit but does not fully meet PLOS ONE’s publication criteria as it currently stands. Therefore, we invite you to submit a revised version of the manuscript that addresses the points raised during the review process. As the authors can see, Reviewer 1 is still asking for a minor revision. Thus, I suggest the authors carefully clarify the two little points and resubmit the manuscript for publication.

We look forward to receiving your revised manuscript.

Kind regards,

Vilfredo De Pascalis

Academic Editor

PLOS ONE

Journal Requirements:

Additional Editor Comments:

As the authors can see, Reviewer 1 is still asking for a minor revision. Thus, I suggest the authors carefully clarify the two little points and resubmit the manuscript for publication.

Reviewers' comments:

Reviewer's Responses to Questions

**Comments to the Author**

1. If the authors have adequately addressed your comments raised in a previous round of review and you feel that this manuscript is now acceptable for publication, you may indicate that here to bypass the “Comments to the Author” section, enter your conflict of interest statement in the “Confidential to Editor” section, and submit your "Accept" recommendation.

Reviewer #1: (No Response)

Reviewer #2: (No Response)

2. Is the manuscript technically sound, and do the data support the conclusions?

Reviewer #1: Yes

Reviewer #2: Yes

3. Has the statistical analysis been performed appropriately and rigorously? 

Reviewer #1: Yes

Reviewer #2: Yes

4. Have the authors made all data underlying the findings in their manuscript fully available?

Reviewer #1: Yes

Reviewer #2: Yes

5. Is the manuscript presented in an intelligible fashion and written in standard English?

Reviewer #1: Yes

Reviewer #2: Yes

6. Review Comments to the Author

Reviewer #1: Overall, the authors were nicely responsive to the critiques raised in the original manuscript. I have only two minor interpretive points remaining in considering the GEE results in Study 1 (as reported in Table 3) that deserve further interpretation. I note it is possible the authors did so already and I just missed these points, but I didn’t find these interpretations on my reads of the revised and tracked-changes versions.

1) What does it mean that MaxMot did not influence the number of hard choices on the original EEfRT but was positively associated with the number of button presses in the modified EEfRT? In particular, I wonder if this pattern indicates the relative importance of basic motoric functioning as an influence on the primary outcome variable on the two versions of the task.

2) What does it mean that the Probability x Reward Magnitude interactions were opposite in sign between the original and modified EEfRT? The two outcomes are coded in the same direction, so the differences in these signs point to another potentially important difference in the two versions of the tasks – and which version is more theoretically compatible with the EEfRT’s aims?

Reviewer #2: I'm fine with the current version. They have addressed the concerns I raised. I did flag some missing words and a couple of other issues. But generally I think it is in good shape.

One issue that still concerns me is that the samples for both studies were recruited from a psychologically healthy population, which could lead to serious restriction of range problems on all the measures, including the EEfRT. Also, the first sample is only 120 or so, which suggests that there will be very few people with elevated or clinical levels of depression.

7. PLOS authors have the option to publish the peer review history of their article (what does this mean?). If published, this will include your full peer review and any attached files.

Reviewer #1: **Yes: **Stephen D. Benning

Reviewer #2: No

---

## [Author Response · Author response to Decision Letter 1]

20 Dec 2021

Response to Reviewer 1:

Reviewer #1: Overall, the authors were nicely responsive to the critiques raised in the original manuscript. I have only two minor interpretive points remaining in considering the GEE results in Study 1 (as reported in Table 3) that deserve further interpretation. I note it is possible the authors did so already and I just missed these points, but I didn’t find these interpretations on my reads of the revised and tracked-changes versions.

1) What does it mean that MaxMot did not influence the number of hard choices on the original EEfRT but was positively associated with the number of button presses in the modified EEfRT? In particular, I wonder if this pattern indicates the relative importance of basic motoric functioning as an influence on the primary outcome variable on the two versions of the task.

Our findings suggest that motoric abilities are more strongly related to the number of clicks in the modified EEfRT compared to hard task choices in the original EEfRT. These results indicate that both tasks also differ in terms of physical demands. Therefore, future work using this task should also include a measure of motoric abilities. We adjusted the discussion accordingly (p. 42, 856-868).

2) What does it mean that the Probability x Reward Magnitude interactions were opposite in sign between the original and modified EEfRT? The two outcomes are coded in the same direction, so the differences in these signs point to another potentially important difference in the two versions of the tasks – and which version is more theoretically compatible with the EEfRT’s aims?

Our findings suggest a different interplay of probability of reward attainment and reward magnitude in the two versions of the EEfRT. In the original EEfRT, the effects of reward magnitude on hard task choices are strongest for higher probabilities of reward attainment. In contrast, in the modified EEfRT, effects of reward magnitude on the number of clicks are largest for smaller probabilities of reward attainment. These differential patterns could be attributable to (1) strategy use in the original EEfRT stemming from the fact that hard task choices for low probability trials may lead to less money won overall, and (2) limits to the additional clicks possible in the modified EEfRT for high probability trials (more effort is required for additional clicks when the number of clicks is already high). We adjusted the discussion accordingly (p. 42, 856-868)

Response to Reviewer 2:

Reviewer #2: I'm fine with the current version. They have addressed the concerns I raised. I did flag some missing words and a couple of other issues. But generally I think it is in good shape.

Thank you! we have incorporated your suggestions in the manuscript. for instance, we have (1) corrected several language issues, (2) made it more explicit that our modifications of the EEfRT (number of clicks vs. choice between two tasks) represents a substantial change (see p. 16, 317-318), and (3) discussed the differences between both task versions in a more detailed manner (see p. 42, l. 856-868)

One issue that still concerns me is that the samples for both studies were recruited from a psychologically healthy population, which could lead to serious restriction of range problems on all the measures, including the EEfRT. Also, the first sample is only 120 or so, which suggests that there will be very few people with elevated or clinical levels of depression.

We agree to your concerns, which we already discuss within our manuscript. We added the restriction of range you mentioned to our discussion (p. 43, l. 883-884).

---

## [Editor Report · Decision Letter 2]

10 Jan 2022

Examining the reliability and validity of two versions of the Effort-Expenditure for Rewards Task (EEfRT)

PONE-D-21-08512R2

Dear Dr. Ohmann,

We’re pleased to inform you that your manuscript has been judged scientifically suitable for publication and will be formally accepted for publication once it meets all outstanding technical requirements.

Kind regards,

Vilfredo De Pascalis

Academic Editor

PLOS ONE

Additional Editor Comments (optional):

In the second revision, the authors addressed all the points raised by Reviewer#1 and Reviewere#2. Thus the manuscript is ready for publication.

I compliment the Reviewers and Authors for their excellent work.
---

## [Editor Report · Acceptance letter]

19 Jan 2022

PONE-D-21-08512R2 

Examining the reliability and validity of two versions of the Effort-Expenditure for Rewards Task (EEfRT) 

Dear Dr. Ohmann:

I'm pleased to inform you that your manuscript has been deemed suitable for publication in PLOS ONE. Congratulations! Your manuscript is now with our production department. 

Kind regards, 

on behalf of

Prof. Vilfredo De Pascalis 

Academic Editor

PLOS ONE